# Identifying Macro Causal Effects in C-DMGs over DMGs

**Simon Ferreira**
Sorbonne Université, INSERM,
Institut Pierre Louis d'Epidémiologie
et de Santé Publique,
F75012, Paris, France
simon.ferreira@sorbonne-universite.fr

**Charles K. Assaad**
Sorbonne Université, INSERM,
Institut Pierre Louis d'Epidémiologie
et de Santé Publique,
F75012, Paris, France
charles.assaad@inserm.fr

## Abstract

The do-calculus is a sound and complete tool for identifying causal effects in acyclic directed mixed graphs (ADMGs) induced by structural causal models (SCMs). However, in many real-world applications, especially in high-dimensional settings, constructing a fully specified ADMG is often infeasible. This limitation has led to growing interest in partially specified causal representations, particularly through cluster-directed mixed graphs (C-DMGs), which group variables into clusters and offer a more abstract yet practical view of causal dependencies. While these representations can include cycles, recent work has shown that the do-calculus remains sound and complete for identifying macro-level causal effects in C-DMGs over ADMGs under the assumption that all clusters sizes are greater than 1. Nevertheless, real-world systems often exhibit cyclic causal dynamics at the structural level. To account for this, input-output structural causal models (ioSCMs) have been introduced as a generalization of SCMs that allow for cycles. ioSCMs induce another type of graph structure known as a directed mixed graph (DMG). Analogous to the ADMG setting, one can define C-DMGs over DMGs as high-level representations of causal relations among clusters of variables. In this paper, we prove that, unlike in the ADMG setting, the do-calculus is unconditionally sound and complete for identifying macro causal effects in C-DMGs over DMGs. Furthermore, we show that the graphical criteria for non-identifiability of macro causal effects previously established C-DMGs over ADMGs naturally extends to a subset of C-DMGs over DMGs.

## 1  Introduction

Understanding and identifying causal effects is a central goal in many scientific disciplines. In recent years, structural causal models (SCMs) have emerged as a foundational framework for reasoning about causality. These models encode causal assumptions through structural equations and are typically represented by acyclic directed mixed graphs (ADMGs), which capture both causal and confounding relationships. Within this framework, the do-calculus [Pearl, 1995]—based on the notion of d-separation [Pearl, 1988]—provides a complete and sound set of inference rules for identifying causal effects from observational data, assuming the causal structure is fully specified. However, SCMs do not fully capture systems with cyclic causal dependencies at the structural level, which are common in public health, biology, economics, and engineering systems. For example, there can be a cyclic relation between poor mental health (*e.g.*, depression or anxiety) and substance use (*e.g.*, alcohol, drugs). The worsening of mental health and increase in substance use can occur in tight time-frames (daily or even hourly), especially in high-risk populations. Over time, they may reach a cyclic equilibrium where both reinforce each other without a clear causal ordering. To address

39th Conference on Neural Information Processing Systems (NeurIPS 2025).

this, the notion of input-output structural causal models (ioSCMs) has been proposed [Forré and Mooij, 2020]. These models generalize SCMs by allowing for cycles and induce a new class of graphs known as directed mixed graphs (DMGs) [Richardson, 1997, Forré and Mooij, 2017, 2018, Forré and Mooij, 2020, Boeken and Mooij, 2024], which provide a richer representation of causal structures. Furthermore, Forré and Mooij [2020] introduced an extension of d-separation to DMGs, called $\sigma$-separation, and showed that the do-calculus, when replacing d-separation by $\sigma$-separation becomes sound for identifying causal effects in DMGs [Forré and Mooij, 2020].

However, in many real-world applications—particularly those involving high-dimensional data or limited domain knowledge—it is often unrealistic to assume a complete specification of the underlying causal graph. This has motivated the development of partially specified graphical models [Maathuis and Colombo, 2013, Perkovic et al., 2016, Perkovic, 2020, Jaber et al., 2022, Wang et al., 2023, Assaad et al., 2023, Anand et al., 2023, Wahl et al., 2024, Reiter et al., 2024, Boeken and Mooij, 2024, Ferreira and Assaad, 2024, 2025a], and in particular cluster graphs. Cluster graphs abstract away some of the fine-grained details by grouping variables into clusters, thus offering a more flexible and scalable representation of complex systems. Importantly, cluster graphs allow for cycles, which can arise naturally in feedback systems or time-dependent processes, complicating the analysis compared to traditional ADMGs. In these graphs, causal effects can be separated into two types: a micro causal effect where the interest is the effect of variable within a cluster on another variable in another cluster; and the macro causal effect where the interest in the effect of a set of an entire cluster on another entire cluster. In this work, we focus on the latter. Anand et al. [2023], Tikka et al. [2023] have shown that do-calculus (the version using d-separation) remains both sound and complete for identifying macro causal effects when the cluster graph representing ADMGs is acyclic. Ferreira and Assaad [2025a,b] showed that the do-calculus (the version using d-separation) is also sound and complete for identifying macro-level causal effects when the cluster graph representing an ADMG is cyclic, denoted here as C-DMG over ADMGs, assuming either that the size of the clusters is unknown, or that each cluster contains more than one variable [Ferreira and Assaad, 2025b].

Motivated by these developments, we consider the problem of identifying macro-level causal effects in cluster graphs representing DMGs, denoted as C-DMGs over DMGs, a natural generalization of previous work. Our contributions are threefold:

- We prove that $\sigma$-separation [Forré and Mooij, 2018]—a fundamental tool in causal reasoning in DMGs—is sound and complete in C-DMGs over DMGs.
- We prove that do-calculus (the version using $\sigma$-separation) [Forré and Mooij, 2020] is sound and complete for identifying macro-level causal effects in C-DMGs over DMGs—unconditionally, and without the constraints needed in the case of C-DMGs over ADMGs [Ferreira and Assaad, 2025b, Yvernes, 2025].
- We show that the graphical characterization of non-identifiability previously developed for C-DMGs over ADMGs [Ferreira and Assaad, 2025a,b] also applies for C-DMGs over DMGs under an additional assumption.

The remainder of the paper is organized as follows: In Section 2, we formally presents C-DMGs over DMGs. In Section 3, we show that $\sigma$-separation and the do-calculus is sound and complete for macro causal effects in C-DMGs over DMGs and present a graphical characterization for the non-identifiability of these effects. Finally in Section 4, we conclude the paper while showing its limitations. All proofs are deferred to the appendix.

## 2 Preliminaries

To streamline the presentation and avoid repetitive explanations, we will adopt the unified notation $\mathcal{G}^* = (\mathbb{V}^*, \mathbb{E}^*)$ to refer to any type of graph. This notation allows us to generalize results and discussions without redundancy across different graph types. In the remainder, for every vertex $V^* \in \mathbb{V}^*$ in a graph $\mathcal{G}^* = (\mathbb{V}^*, \mathbb{E}^*)$, we will refer to its parents by $Pa(V^*, \mathcal{G}^*)$, its ancestors by $An(V^*, \mathcal{G}^*)$, and its descendants by $De(V^*, \mathcal{G}^*)$. We consider that a vertex counts as its own descendant and as its own ancestor. In addition, the strongly connected component of a vertex is defined as $Scc(V, \mathcal{G}^*) = An(V, \mathcal{G}^*) \cap De(V, \mathcal{G}^*)$.

In this section, we present the essential definitions and notations that will be used throughout the paper, ensuring clarity and consistency in the exposition of our results. In this work, we assume

causal relations are modeled using an input/output structural causal model (ioSCM) [Forré and Mooij, 2020]—which extends classical structural causal models (SCMs) [Pearl, 2009] by allowing for the presence of cycles. Unlike classical SCMs, ioSCMs allow structural equations to mutually depend on each other. For example, in the cyclic system:

$$X \coloneqq f_X(Y, L_X) \qquad ; \qquad Y \coloneqq f_Y(X, L_Y) \qquad ; \qquad (X, Y) \coloneqq f_{(X,Y)}(L_X, L_Y),$$

$X$ functionally depends on $Y$, and $Y$ functionally depends on $X$, forming cycle. When cycles exist, instead of computing variables in a top-down order as in SCMs, ioSCMs rely on fixed-point solutions. That is, a joint assignment to the variables that simultaneously satisfies all equations. This is analogous to finding an equilibrium in dynamic systems. Firstly, let us properly define the notion of loops as it will be useful to guarantee the compatibility of the causal mechanisms in ioSCMs.

**Definition 1** (Loops). *In a directed graph $\mathcal{G}^* = (\mathbb{V}^*, \mathbb{E}^*)$, a loop is a set of vertices $\mathbb{S} \subseteq \mathbb{V}^*$ such that there exists a directed path between every pair of distinct vertices in the subgraph induced by $\mathbb{S}$ i.e., $\forall U \neq V \in \mathbb{S}, V \in De(U, \mathcal{G}^*|_{\mathbb{S}})$.*

*Note that every singleton $\mathbb{S} \in \{\{V\} \mid V \in \mathbb{V}^*\}$ and every strongly connected components $\mathbb{S} \in \{Scc(V, \mathcal{G}^*) \mid V \in \mathbb{V}^*\}$ are loops. We write the set of all loops of the graph $\mathcal{G}^*$ as $\mathcal{L}(\mathcal{G}^*)$. We call cycles the loops that are not singletons.*

Next, we recall the definition of ioSCMs from Forré and Mooij [2020] with the omission of the domains of the variables.

**Definition 2** (input/output Structural Causal Model (ioSCM)). *An input/output structural causal model is a tuple $\mathcal{M} = (\mathbb{L}, \mathbb{V}_{obs}, \mathbb{J}, \mathcal{G}^+, \mathbb{F}, \Pr(\mathbb{I}))$, where*

- *$\mathbb{L}$ is a set of latent/exogenous variables, which cannot be observed but affect the rest of the model.*

- *$\mathbb{V}_{obs}$ is a set of observed/endogenous variables, which are observed and every $V \in \mathbb{V}_{obs}$ is functionally dependent on some subset of $(\mathbb{L} \cup \mathbb{V}_{obs} \cup \mathbb{J}) \setminus \{V\}$.*

- *$\mathbb{J}$ is a set of input/intervention variables which are not functionally dependent of any other variable but rather are fixed to specific values.*

- *$\mathcal{G}^+ = (\mathbb{V}^+, \mathbb{E}^+)$ is a graphical structure where:*

    - *$\mathbb{V}^+ = \mathbb{V}_{obs} \cup \mathbb{L} \cup \mathbb{J}$*
    - *$\mathbb{V}_{obs} = Ch(\mathbb{L} \cup \mathbb{J}, \mathcal{G}^+)$*
    - *$Pa(\mathbb{L} \cup \mathbb{J}, \mathcal{G}^+) = \varnothing$*

- *$\mathbb{F}$ is a set of functions such that for all $\mathbb{S} \in \mathcal{L}(\mathcal{G}^+|_{\mathbb{V}_{obs}})$, $f^{\mathbb{S}}$ is a function taking as input the values of $Pa(\mathbb{S}, \mathcal{G}^+) \setminus \mathbb{S}$ and outputting values for $\mathbb{S}$ and such that $\mathbb{F}$ satisfies the global compatibility condition:*

$$
\begin{aligned}
\forall \mathbb{S}' \subsetneqq \mathbb{S} &\in \mathcal{L}(\mathcal{G}^+|_{\mathbb{V}_{obs}}), \forall \mathbb{v} \text{ values of } Pa(\mathbb{S}, \mathcal{G}^+) \cup \mathbb{S}, \\
f^{\mathbb{S}}(\mathbb{v}|_{Pa(\mathbb{S}, \mathcal{G}^+) \setminus \mathbb{S}}) &= \mathbb{v}|_{\mathbb{S}} \implies f^{\mathbb{S}'}(\mathbb{v}|_{Pa(\mathbb{S}', \mathcal{G}^+) \setminus \mathbb{S}'}) = \mathbb{v}|_{\mathbb{S}'}
\end{aligned} \tag{1}
$$

- *$\Pr(\mathbb{I})$ is a joint probability distribution over $\mathbb{L}$.*

An ioSCM induces a directed graph, where every variable in $\mathbb{V}^+$ corresponds to a vertex in the graph. In this directed graph, a directed edge $\rightarrow$ is drawn from one variable to another if the former serves as an input to the function that determines the latter. For simplicity, instead of working directly with these directed graphs, we consider an alternative representation known as a directed mixed graph (DMG). In a DMG, only the observed and intervened variables (*i.e.*, $\mathbb{V}_{obs} \cup \mathbb{J}$) correspond to vertices, while hidden variables in $\mathbb{L}$ that share common outputs are represented by bidirected edges $\leftarrow\dashrightarrow$ between the corresponding observed variables, thereby implicitly accounting for the hidden confounding. Formally, DMGs are defined as follows:

**Definition 3** (Directed mixed graph (DMG)). *Consider an ioSCM $\mathcal{M}$. The directed mixed graph $\mathcal{G} = (\mathbb{V}, \mathbb{E} = \mathbb{E}_{\rightarrow} \cup \mathbb{E}_{\leftarrow\dashrightarrow})$ induced by $\mathcal{M}$ is the DMG where:*

- *the vertices $\mathbb{V} = \mathbb{V}_{obs} \cup \mathbb{J}$ are the endogenous variables and the intervention variables of the ioSCM; and*

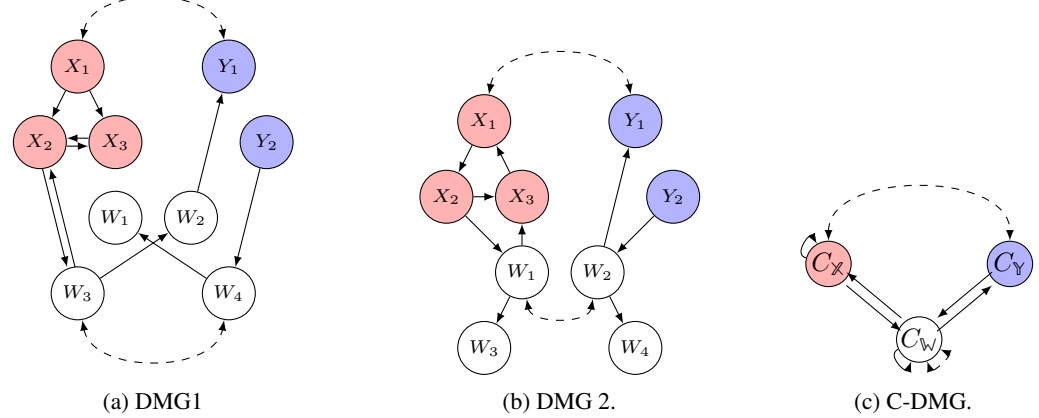

(a) DMG1       (b) DMG 2.       (c) C-DMG.

Figure 1: Two DMGs and their compatible C-DMG. Red vertices represent the exposures of interest in and blue vertices represent the outcome of interest.

- *the directed edges in $\mathcal{G}$ are $\mathbb{E}_\rightarrow = \mathbb{E}^+|_{\mathbb{V}_{obs} \cup \mathbb{J}}$; and*
- *the bidirected edges in $\mathcal{G}$ are $\mathbb{E}_{\leftarrow\cdots\rightarrow} = \{X \leftarrow\cdots\rightarrow Y \mid X, Y \in \mathbb{V}, \exists L \in \mathbb{L} \text{ such that } L \rightarrow X, L \rightarrow Y \in \mathbb{E}^+\}.*

However, in many fields, constructing, analyzing, and validating a DMG remains a significant challenge for researchers due to the inherent difficulty in accurately determining causal relationships among individual variables. This complexity primarily stems from the uncertainty surrounding causal relations, making it challenging to specify the precise structure of the graph. Nevertheless, researchers can often provide a partially specified version of the DMG, which offers a more practical and compact representation of the underlying causal structure. These simplified representations, which we call Cluster-Directed Mixed Graphs over DMGs (C-DMGs over DMGs), group several variables into clusters, allowing for the representation of causal relationships at a higher level of abstraction while retaining essential structural properties of the system. In a C-DMG over DMGs, directed edges between clusters represent causal influences at the higher level, while bidirected edges capture hidden confounding effects that exist between clusters. Formally, C-DMGs over DMGs are defined as follows:

**Definition 4** (Cluster directed mixed graph over DMGs (C-DMG over DMGs)). *Let $\mathcal{G} = (\mathbb{V}, \mathbb{E})$ be a DMG induced from an ioSCM $\mathcal{M}$ and $\mathbb{C} = \{C_1, \cdots, C_k\}$ a partition of $\mathbb{V}$. A C-DMG over DMGs compatible with $\mathcal{G}$ according to $\mathbb{C}$ is a graph $\mathcal{G}^\mathbb{C} = (\mathbb{C}, \mathbb{E}^\mathbb{C})$ where $\forall C_i, C_j \in \mathbb{C}$ the edge $C_i \rightarrow C_j$ (resp. $C_i \leftarrow\cdots\rightarrow C_j$) is in $\mathbb{E}^\mathbb{C}$ if and only if there exists $V_i \in C_i$ and $V_j \in C_j$ such that $V_i \rightarrow V_j$ (resp. $V_i \leftarrow\cdots\rightarrow V_j$) is in $\mathbb{E}$.*

Figure 1 presents a simple C-DMG over DMGs along with two of its compatible DMGs. Cycles in a C-DMG over DMGs can arise for two distinct reasons. First, unlike in a C-DMG over ADMGs, a C-DMG over DMGs can contain a cycle if there is a genuine cyclic relationship in the underlying DMG between nodes belonging to different clusters. For example, in Figure 1a the cycle between $X_2$ and $W_3$ in the DMG induces a cycle between clusters $C_\mathbb{X}$ and $C_\mathbb{W}$ in the corresponding C-DMG over DMGs in Figure 1c. Secondly, even in the absence of an actual cycle in the underlying DMG, cycles can appear in the C-DMG over DMGs due to its partial specification. This is illustrated in Figure 1b, where the edges $X_2 \rightarrow W_1$ and $W_1 \rightarrow X_3$ together create a cycle between clusters $C_\mathbb{X}$ and $C_\mathbb{W}$ in the C-DMG over DMGs in Figure 1c. Lastly, cycles that are contained in a single cluster do not appear in the C-DMG over DMGs. This is illustrated in Figure 1a with the cycle $X_2 \rightleftarrows X_3$ that does not show in the C-DMG over DMGs in Figure 1c.

We distinguish between two types of causal effects in the context of C-DMGs, the macro causal effect [Anand et al., 2023, Ferreira and Assaad, 2025a,b] and the micro causal effect [Assaad et al., 2024, Assaad, 2025]. In this paper we focus on the former and we formally define it below:

**Definition 5** (Macro causal effect). *Consider a DMG $\mathcal{G}$ over variables $\mathbb{V}$ induced from an ioSCM and let $\mathcal{G}^\mathbb{C} = (\mathbb{C}, \mathbb{E}^\mathbb{C})$ be a compatible C-DMG. A macro causal effect is a causal effect from a set of macro-variables $\mathbb{C}_\mathbb{X}$ on another set of macro-variables $\mathbb{C}_\mathbb{Y}$ where $\mathbb{C}_\mathbb{X}$ and $\mathbb{C}_\mathbb{Y}$ are disjoint subsets*

*of* $\mathbb{C}$. *It is written* $\Pr\left(\mathbb{C}_{\mathbb{Y}} = c_{\mathbb{Y}} \mid do\left(\mathbb{C}_{\mathbb{X}} = c_{\mathbb{X}}\right)\right)$*, where the* $do\left(\cdot\right)$ *operator represents an external intervention.*

The identification problem in causal inference aims to establish whether a causal effect of a set of variables on another set of variables can be expressed exclusively in terms of observed variables and standard probabilistic notions, such as conditional probabilities. Formally, the identification problem in the context of macro causal effects and C-DMGs over DMGs is defined as follows:

**Definition 6** (Identifiability in C-DMGs over DMGs)**.** *Let* $\mathbb{C}_{\mathbb{X}}$ *and* $\mathbb{C}_{\mathbb{Y}}$ *be disjoint sets of vertices in a C-DMGs over DMGs* $\mathcal{G}^{\mathbb{c}}$. *The macro causal effect of* $\mathbb{C}_{\mathbb{X}}$ *on* $\mathbb{C}_{\mathbb{Y}}$ *is identifiable in* $\mathcal{G}^{\mathbb{c}}$ *if* $\Pr\left(\mathbb{C}_{\mathbb{Y}} = c_{\mathbb{Y}} \mid do\left(\mathbb{C}_{\mathbb{X}} = c_{\mathbb{X}}\right)\right)$ *is uniquely computable from any observational positive distribution compatible with* $\mathcal{G}^{\mathbb{c}}$.

In the following, we will abuse the notation by writing $\Pr\left(c_{\mathbb{Y}} \mid do\left(c_{\mathbb{X}}\right)\right)$ instead of $\Pr\left(\mathbb{C}_{\mathbb{Y}} = c_{\mathbb{Y}} \mid do\left(\mathbb{C}_{\mathbb{X}} = c_{\mathbb{X}}\right)\right)$ when the setting is clear. In addition, whenever the context is clear, we will refer to C-DMGs over DMGs simply as C-DMGs.

# 3 Identification of Macro Causal Effects in C-DMGs over DMGs

In this section, we aim to establish that the do-calculus is both sound and complete for identifying macro-level causal effects in a C-DMG over DMGs. We begin by showing, in the first subsection, that $\sigma$-separation—originally developed for DMGs as a tool for identifying conditional independencies—remains sound and complete when extended to C-DMGs over DMGs for detecting macro-level conditional independencies. In the second subsection, we present the core theoretical contribution of this section: the soundness and completeness of do-calculus for macro causal effect identification in this setting. Finally, we provide a graphical characterization of non-identifiability, shedding light on cases where causal effects cannot be inferred from observational data alone.

## 3.1 The $\sigma$-separation in C-DMGs over DMGs

The standard notion of d-separation [Pearl, 1988] was originally introduced for acyclic directed mixed graphs (ADMGs). It was later shown to remain valid when extended to C-ADMGs over ADMGs [Anand et al., 2023] and C-DMGs over ADMGs [Ferreira and Assaad, 2025a,b]. However, d-separation does not apply to DMGs, which may contain cyclic causal relations in the SCM. To address this limitation, $\sigma$-separation was introduced as a generalization suitable for DMGs [Forré and Mooij, 2020]. In this subsection, we demonstrate that $\sigma$-separation can be naturally applied to C-DMGs over DMGs. We begin by formally defining $\sigma$-blocked walks and the concept of $\sigma$-separation in this generalized setting.

**Definition 7** ($\sigma$-blocked walk [Forré and Mooij, 2020])**.** *In a graph* $\mathcal{G}^* = \left(\mathbb{V}^*, \mathbb{E}^*\right)$*, a walk* $\tilde{\pi} = \left\langle V_1^*, \cdots, V_n^* \right\rangle$ *is said to be* $\sigma$*-blocked by a set of vertices* $\mathbb{W}^* \subseteq \mathbb{V}^*$ *if:*

1. $V_1^* \in \mathbb{W}^*$ *or* $V_n^* \in \mathbb{W}^*$, *or*

2. $\exists 1 < i < n$ *such that* $\left\langle V_{i-1}^* \leftrightarrowtail V_i^* \leftarrowtail V_{i+1}^* \right\rangle \subseteq \tilde{\pi}$ *and* $V_i^* \notin \mathbb{W}^*$, *or*

3. $\exists 1 < i < n$ *such that* $\left\langle V_{i-1}^* \leftarrow V_i^* \leftarrowtail V_{i+1}^* \right\rangle \subseteq \tilde{\pi}$ *and* $V_i^* \in \mathbb{W}^* \backslash Scc\left(V_{i-1}^*, \mathcal{G}^*\right)$, *or*

4. $\exists 1 < i < n$ *such that* $\left\langle V_{i-1}^* \leftrightarrowtail V_i^* \rightarrow V_{i+1}^* \right\rangle \subseteq \tilde{\pi}$ *and* $V_i^* \in \mathbb{W}^* \backslash Scc\left(V_{i+1}^*, \mathcal{G}^*\right)$, *or*

5. $\exists 1 < i < n$ *such that* $\left\langle V_{i-1}^* \leftarrow V_i^* \rightarrow V_{i+1}^* \right\rangle \subseteq \tilde{\pi}$ *and*
   $V_i^* \in \mathbb{W}^* \backslash \left(Scc\left(V_{i-1}^*, \mathcal{G}^*\right) \cap Scc\left(V_{i+1}^*, \mathcal{G}^*\right)\right)$.

*where* $\leftrightarrowtail$ *represents* $\rightarrow$ *or* $\leftarrow\!\cdots\!\rightarrow$*,* $\leftarrowtail$ *represents* $\leftarrow$ *or* $\leftarrow\!\cdots\!\rightarrow$*, and* $\leftrightarrowtail\!\!\!\!\!\rightarrow$ *represents any of the three arrow type* $\rightarrow$*,* $\leftarrow$ *or* $\leftarrow\!\cdots\!\rightarrow$*. A walk which is not* $\sigma$*-blocked is said to be* $\sigma$*-active.*

**Definition 8** ($\sigma$-separation [Forré and Mooij, 2020])**.** *In a graph* $\mathcal{G}^* = \left(\mathbb{V}^*, \mathbb{E}^*\right)$*, let* $\mathbb{X}^*, \mathbb{Y}^*, \mathbb{W}^*$ *be distinct subsets of* $\mathbb{V}^*$*.* $\mathbb{W}^*$ *is said to* $\sigma$*-separate* $\mathbb{X}^*$ *and* $\mathbb{Y}^*$ *if and only if* $\mathbb{W}^*$ $\sigma$*-blocks every walk from a vertex in* $\mathbb{X}^*$ *to a vertex in* $\mathbb{Y}^*$*. It is written* $\left(\mathbb{X}^* \perp\!\!\!\perp_\sigma \mathbb{Y}^* \mid \mathbb{W}^*\right)_{\mathcal{G}^*}$*.*

The following theorem shows that $\sigma$-separation is applicable as is to C-DMGs over DMGs.

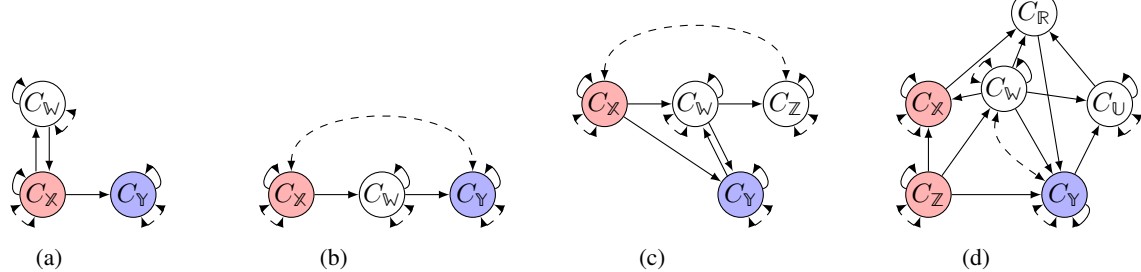

Figure 2: C-DMGs with identifiable macro causal effects. Each pair of red and blue vertices represents the causal effect we are interested in.

**Theorem 1** (Soundness of $\sigma$-separation in C-DMGs over DMGs). *Let $\mathcal{G}^{\mathfrak{c}} = (\mathbb{C}, \mathbb{E}^{\mathfrak{c}})$ be a C-DMG and $\mathbb{C}_{\mathbb{X}}, \mathbb{C}_{\mathbb{Y}}, \mathbb{C}_{\mathbb{W}}$ be disjoint subsets of $\mathbb{C}$. If $\mathbb{C}_{\mathbb{X}}$ and $\mathbb{C}_{\mathbb{Y}}$ are $\sigma$-separated by $\mathbb{C}_{\mathbb{W}}$ in $\mathcal{G}^{\mathfrak{c}}$ then, in any compatible DMG $\mathcal{G} = (\mathbb{V}, \mathbb{E})$, $\mathbb{X} = \bigcup_{C \in \mathbb{C}_{\mathbb{X}}} C$ and $\mathbb{Y} = \bigcup_{C \in \mathbb{C}_{\mathbb{Y}}} C$ are $\sigma$-separated by $\mathbb{W} = \bigcup_{C \in \mathbb{C}_{\mathbb{W}}} C$.*

Theorem 1 establishes that $\sigma$-separation in C-DMGs over DMGs ensures the existence of a corresponding macro-level $\sigma$-separation across all compatible DMGs. According to [Forré and Mooij, 2020, Theorem 5.2], this implies that some conditional independencies in the underlying probability distribution can be inferred directly from the C-DMG. By extending the applicability of $\sigma$-separation to C-DMGs over DMGs, this result enables the identification of macro-level conditional independencies even when the underlying causal structure is only partially specified. To illustrate the practical value of this result, we now present two examples demonstrating the application of $\sigma$-separation in a C-DMG over DMGs.

**Example 1.** *Let $\mathcal{G}$ be the true unknown DMG and consider that its compatible C-DMG, denoted as $\mathcal{G}^{\mathfrak{c}}$ is one given in Figure 2a. Using Definition 8, we can directly deduce $(C_{\mathbb{W}} \perp\!\!\!\perp_{\sigma} C_{\mathbb{Y}} \mid C_{\mathbb{X}})_{\mathcal{G}^{\mathfrak{c}}}$. Thus according to Theorem 1 and [Forré and Mooij, 2020, Theorem 5.2], $C_{\mathbb{W}}$ is conditionally independent of $C_{\mathbb{Y}}$ given $C_{\mathbb{X}}$ in every distribution compatible with the true ADMG.*

**Example 2.** *Let $\mathcal{G}$ be the true unknown DMG and consider that its compatible C-DMG, denoted as $\mathcal{G}^{\mathfrak{c}}$ is one given in Figure 2c. Using Definition 8, we can directly deduce that $(C_{\mathbb{Z}} \perp\!\!\!\perp_{\sigma} C_{\mathbb{Y}} \mid C_{\mathbb{X}}, C_{\mathbb{W}})_{\mathcal{G}^{\mathfrak{c}}}$. Thus according to Theorem 1 and [Forré and Mooij, 2020, Theorem 5.2], $C_{\mathbb{Z}}$ is conditionally independent of $C_{\mathbb{Y}}$ given $C_{\mathbb{X}}$ and $C_{\mathbb{W}}$ in every distribution compatible with the true ADMG.*

The following theorem shows that $\sigma$-separation is also complete in C-DMGs over DMGs.

**Theorem 2** (Completeness of $\sigma$-separation in C-DMGs). *Let $\mathcal{G}^{\mathfrak{c}} = (\mathbb{C}, \mathbb{E}^{\mathfrak{c}})$ be a C-DMG, $\mathbb{C}_{\mathbb{X}}, \mathbb{C}_{\mathbb{Y}}, \mathbb{C}_{\mathbb{W}}$ be disjoint subsets of $\mathbb{C}$, $\mathbb{X} = \bigcup_{C \in \mathbb{C}_{\mathbb{X}}} C$, $\mathbb{Y} = \bigcup_{C \in \mathbb{C}_{\mathbb{Y}}} C$ and $\mathbb{W} = \bigcup_{C \in \mathbb{C}_{\mathbb{W}}} C$. If $\mathbb{C}_{\mathbb{X}}$ and $\mathbb{C}_{\mathbb{Y}}$ are not $\sigma$-separated by $\mathbb{C}_{\mathbb{W}}$ in $\mathcal{G}^{\mathfrak{c}}$, then there exists a compatible DMG $\mathcal{G} = (\mathbb{V}, \mathbb{E})$ such that $\mathbb{X}$ and $\mathbb{Y}$ are not $\sigma$-separated by $\mathbb{W}$.*

The findings of Theorems 1 and 2 establish that identifying a $\sigma$-separation in C-DMGs over DMGs ensures the recovery of all common macro-level $\sigma$-separations across all compatible DMGs. This result is particularly valuable for constraint-based causal discovery methods, especially when the goal is to infer the structure of a C-DMG without needing to fully specify an underlying DMG. Most importantly, these insights lay the theoretical groundwork for the results developed in the next subsection.

## 3.2 The do-calculus in C-DMGs over DMGs

[Pearl, 1995] introduced an important tool in causal reasoning referred to as the do-calculus. This do-calculus consists of three rules each relying on some d-separation in the ADMG to guarantee an equality between different probabilities. Every one of these three rules can be interpreted differently: the first one allows the insertion or deletion of an observation, the second one allows for the exchange between actions and observations and the third one allows the insertion or deletion of actions. The do-calculus in ADMGs is complete and thus allows, whenever it is possible, to identify causal effects. In other words, it allows, whenever it is possible, to express a causal effect containing a do $(\cdot)$ operator as a probabilistic expression without any do $(\cdot)$ operator and thus allows to compute it from a positive observational distribution. Since the do-calculus was initially introduced for ADMGs, it is not easily

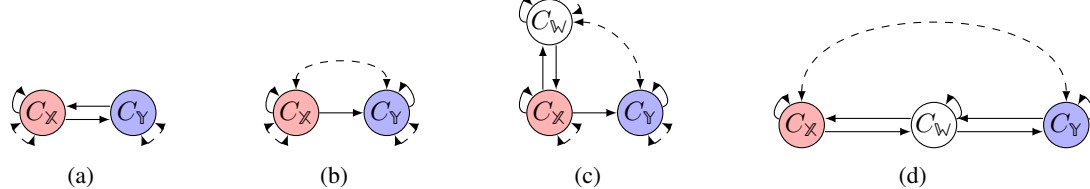

Figure 3: C-DMGs with not identifiable macro causal effects. Each pair of red and blue vertices represents the total effect we are interested in.

extendable to cyclic graphs. However, Forré and Mooij [2020] showed that replacing d-separation by $\sigma$-separation in the three rules allows the do-calculus to be applicable on DMGs induced by ioSCMs. In this subsection, we show that this version of the do-calculus is also readily applicable to C-DMGs over DMGs.

There exists multiple equivalent ways of writing the do-calculus rules, for example [Pearl, 2009] uses the notion of mutilated graph to define the three rules of do-calculus. In this paper, we will follow the notation used in Forré and Mooij [2020]. To do so, we will extend every graph $\mathcal{G}^* = (\mathbb{V}^*, \mathbb{E}^*)$, by adding an intervention vertex $I_X$ and the edge $I_X \rightarrow X$ for every vertex of the graph $X \in \mathbb{V}^*$. Moreover, we will use the $\sigma$-separation notation with a do $(\cdot)$ operator (*e.g.*, $(A \perp\!\!\!\perp_\sigma B \mid C, \mathrm{do}\,(D))_{\mathcal{G}^*}$) to place ourselves in the intervened graph *i.e.*, where all arrows going in $D$ are deleted and in which $D$ is conditioned on[1]. More formal definitions of extended graphs and intervened graphs can be found in the appendix. Using these newly defined notations and $\sigma$-separation we can now state the rules of do-calculus and show their applicability to C-DMGs over DMGs.

**Theorem 3** (do-calculus for C-DMGs over DMGs and macro causal effects). *Let $\mathcal{G}^{\mathbb{c}} = (\mathbb{C}, \mathbb{E}^{\mathbb{c}})$ be a C-DMG over DMGs and $\mathbb{C}_{\mathbb{X}}, \mathbb{C}_{\mathbb{Y}}, \mathbb{C}_{\mathbb{Z}}, \mathbb{C}_{\mathbb{W}}$ be disjoint subsets of $\mathbb{C}$. The three following rules of the do-calculus are sound.*

> ***Rule 1:*** $\Pr\left(\mathbb{c}_{\mathbb{y}} \mid do\left(\mathbb{c}_{\mathbb{z}}\right), \mathbb{c}_{\mathbb{x}}, \mathbb{c}_{\mathbb{w}}\right) = \Pr\left(\mathbb{c}_{\mathbb{y}} \mid do\left(\mathbb{c}_{\mathbb{z}}\right), \mathbb{c}_{\mathbb{w}}\right)$
> *if* $\left(\mathbb{C}_{\mathbb{Y}} \perp\!\!\!\perp_\sigma \mathbb{C}_{\mathbb{X}} \mid \mathbb{C}_{\mathbb{W}}, do\left(\mathbb{C}_{\mathbb{Z}}\right)\right)_{\mathcal{G}^{\mathbb{c}}}$
>
> ***Rule 2:*** $\Pr\left(\mathbb{c}_{\mathbb{y}} \mid do\left(\mathbb{c}_{\mathbb{z}}\right), do\left(\mathbb{c}_{\mathbb{x}}\right), \mathbb{c}_{\mathbb{w}}\right) = \Pr\left(\mathbb{c}_{\mathbb{y}} \mid do\left(\mathbb{c}_{\mathbb{z}}\right), \mathbb{c}_{\mathbb{x}}, \mathbb{c}_{\mathbb{w}}\right)$
> *if* $\left(\mathbb{C}_{\mathbb{Y}} \perp\!\!\!\perp_\sigma \mathbb{I}_{\mathbb{C}_{\mathbb{X}}} \mid \mathbb{C}_{\mathbb{X}}, \mathbb{C}_{\mathbb{W}}, do\left(\mathbb{C}_{\mathbb{Z}}\right)\right)_{\mathcal{G}^{\mathbb{c}}}$
>
> ***Rule 3:*** $\Pr\left(\mathbb{c}_{\mathbb{y}} \mid do\left(\mathbb{c}_{\mathbb{z}}\right), do\left(\mathbb{c}_{\mathbb{x}}\right), \mathbb{c}_{\mathbb{w}}\right) = \Pr\left(\mathbb{c}_{\mathbb{y}} \mid do\left(\mathbb{c}_{\mathbb{z}}\right), \mathbb{c}_{\mathbb{w}}\right)$
> *if* $\left(\mathbb{C}_{\mathbb{Y}} \perp\!\!\!\perp_\sigma \mathbb{I}_{\mathbb{C}_{\mathbb{X}}} \mid \mathbb{C}_{\mathbb{W}}, do\left(\mathbb{C}_{\mathbb{Z}}\right)\right)_{\mathcal{G}^{\mathbb{c}}}$

Now that the rules of do-calculus have been stated for C-DMGs over DMGs and their soundness proven, one can use them sequentially to identify the causal effect $\Pr\left(\mathbb{c}_{\mathbb{y}} \mid \mathrm{do}\left(\mathbb{c}_{\mathbb{x}}\right)\right)$ in all C-DMGs over DMGs in Figure 2. That is to say, to express the causal effect as a probabilistic expression without any do $(\cdot)$ operator. This is done in the following examples.

**Example 3.** *Both in Figure 2a and 2c, one can verify that $(C_{\mathbb{Y}} \perp\!\!\!\perp_\sigma I_{C_{\mathbb{X}}} \mid C_{\mathbb{X}})_{\mathcal{G}^{\mathbb{c}}}$, thus Rule 2 of the do-calculus is applicable and $\Pr\left(c_{\mathbb{y}} \mid do\left(c_{\mathbb{x}}\right)\right) = \Pr\left(c_{\mathbb{y}} \mid c_{\mathbb{x}}\right)$.*

**Example 4.** *Notice that Figure 2b does not contain any cycle other than self-loops and is very similar to Figure 1(b) of Anand et al. [2023] which corresponds to the well-known front-door criterion [Pearl, 2009]. [Forré and Mooij, 2018] have shown that in the acyclic case, $\sigma$-separation coincides with d-separation. Thus, using the corresponding sequence of classical rules of probability and rules of do-calculus as the one given in [Pearl, 2009, p.83], one obtains $\Pr\left(c_{\mathbb{y}} \mid do\left(c_{\mathbb{x}}\right)\right) = \sum_{c_{\mathbb{w}}} \Pr\left(c_{\mathbb{w}} \mid c_{\mathbb{x}}\right) \sum_{c_{\mathbb{x}'}} \Pr\left(c_{\mathbb{y}} \mid c_{\mathbb{w}}, c_{\mathbb{x}'}\right) \Pr\left(c_{\mathbb{x}'}\right)$.*

**Example 5.** *Consider the C-DMG over DMGs in Figure 2d containing a cycle between $C_{\mathbb{Y}}, C_{\mathbb{R}},$ and $C_{\mathbb{U}}$ and a hidden confounding between $C_{\mathbb{Y}}$ and $C_{\mathbb{W}}$. Let $\Pr\left(c_{\mathbb{y}} \mid do\left(c_{\mathbb{x}}, c_{\mathbb{z}}\right)\right)$ be the causal effect of interest. Using the rule of total probability we can rewrite $\Pr\left(c_{\mathbb{y}} \mid do\left(c_{\mathbb{x}}, c_{\mathbb{z}}\right)\right)$ as*

$$\sum_{c_{\mathbb{w}}} \Pr\left(c_{\mathbb{y}} \mid do\left(c_{\mathbb{x}}, c_{\mathbb{z}}\right), c_{\mathbb{w}}\right) \Pr\left(c_{\mathbb{w}} \mid do\left(c_{\mathbb{x}}, c_{\mathbb{z}}\right)\right).$$

---

[1]In other words we write $(A \perp\!\!\!\perp_\sigma B \mid C, \mathrm{do}\,(D))_{\mathcal{G}^*}$ to mean, in Pearl [2009]'s notation, $(A \perp\!\!\!\perp_\sigma B \mid C, D)_{\mathcal{G}^*_{\overline{D}}}$ where $\mathcal{G}^*_{\overline{D}}$ is obtained from $\mathcal{G}^*$ by removing every edge going in $D$.

We first focus on $\Pr\left(c_{\mathbb{Y}} \mid do\left(c_{\mathbb{X}}, c_{\mathbb{Z}}\right), c_{\mathbb{W}}\right)$. Notice that $\left(C_{\mathbb{Y}} \perp\!\!\!\perp_\sigma I_{C_{\mathbb{X}}} \mid C_{\mathbb{X}}, C_{\mathbb{W}}, do\left(C_{\mathbb{Z}}\right)\right)_{\mathcal{G}^c}$ and that $\left(C_{\mathbb{Y}} \perp\!\!\!\perp_\sigma I_{C_{\mathbb{Z}}} \mid C_{\mathbb{Z}}, C_{\mathbb{X}}, C_{\mathbb{W}}\right)_{\mathcal{G}^c}$ which means using two consecutive applications of Rule 2 we can rewrite $\Pr\left(c_{\mathbb{Y}} \mid do\left(c_{\mathbb{X}}, c_{\mathbb{Z}}\right), c_{\mathbb{W}}\right)$ as: $\Pr\left(c_{\mathbb{Y}} \mid c_{\mathbb{Z}}, c_{\mathbb{X}}, c_{\mathbb{W}}\right)$.

Now we focus on $\Pr\left(c_{\mathbb{W}} \mid do\left(c_{\mathbb{X}}, c_{\mathbb{Z}}\right)\right)$. Notice that $\left(C_{\mathbb{W}} \perp\!\!\!\perp_\sigma \mathbb{I}_{C_{\mathbb{X}}} \mid do\left(C_{\mathbb{Z}}\right)\right)_{\mathcal{G}^c}$ which means by Rule 3 of the do-calculus we can completely remove $do\left(c_{\mathbb{X}}\right)$ from the expression. Furthermore, we have $\left(C_{\mathbb{W}} \perp\!\!\!\perp_\sigma \mathbb{I}_{C_{\mathbb{Z}}} \mid C_{\mathbb{Z}}\right)_{\mathcal{G}^c}$ which means by Rule 2 we can replace $do\left(c_{\mathbb{Z}}\right)$ by $c_{\mathbb{Z}}$. So we can rewrite $\Pr\left(c_{\mathbb{W}} \mid do\left(c_{\mathbb{X}}, c_{\mathbb{Z}}\right)\right)$ as $\Pr\left(c_{\mathbb{W}} \mid c_{\mathbb{Z}}\right)$.

These three examples show how the rules of do-calculus in C-DMGs on DMGs can be used to write macro causal effects as an expression of observed probabilities. Therefore, the macro causal effects in these C-DMGs can be estimated from the observational data, provided there is no further issues in the data (*e.g.*, positivity violations).

In the following theorem, we show that not only is the do-calculus applicable in C-DMGs over DMGs, but it is also complete.

**Theorem 4** (Completeness of do-calculus for C-DMGs and macro causal effects). *If one of the do-calculus rules does not apply for a given C-DMG over DMGs, then there exists a compatible DMG for which the corresponding rule does not apply.*

The proof of Theorem 4 relies on a specific compatible DMG called the maximal compatible DMG which is properly defined in the appendix, therefore it can be reformulated as: if one of the do-calculus rules does not apply for a given C-DMG over DMGs, then this same rule does not apply for its maximal compatible DMG.

Note that this completeness result links C-DMGs to the underlying compatible DMGs, however it does not guarantee the absence of other rules to identify causal effects. While the do-calculus based on d-separation in ADMGs introduced in [Pearl, 1995] has been proven to be complete [Shpitser and Pearl, 2006, Huang and Valtorta, 2006], the do-calculus based on $\sigma$-separation in DMGs is not yet proven to be complete and thus our results suffer the same limitation. Using the completeness of the do-calculus in C-DMGs (Theorem 4), one can be convinced, by going through every possible sequence of rules of the do-calculus, that the causal effects of interest in every C-DMGs depicted in Figure 1 and 3 is not identifiable. Unfortunately, going through every possible sequence of rules of the do-calculus is time-consuming and can become impractical when considering larger graphs. In an effort to solve this issue, the following subsection introduces a sub-graphical structure which allows to recognize more efficiently when a causal effect is not identifiable using the do-calculus.

### 3.3 Non-Identifiability: a graphical characterization

In ADMGs, there exists a sub-graphical structure, called a hedge [Shpitser and Pearl, 2006], which is employed to graphically characterize non-identifiability as shown in Shpitser and Pearl 2006, Theorem 4. This graphical criterion is complete when considering ADMGs, that is to say, if the total effect of $X$ on $Y$ is not identifiable in an ADMG then there exists a hedge for $(X, Y)$ [Shpitser and Pearl, 2006]. However, it has been shown that this characterization is too weak to characterize all non-identifiabilities in the case of C-DMGs over ADMGs [Ferreira and Assaad, 2025b]. Thus, a modified version of the hedge structure called the SC-hedge (strongly connected hedge) has been introduced in Ferreira and Assaad [2025a]. The presence of such SC-hedge in C-DMGs over ADMGs guarantees non-identifiability [Ferreira and Assaad, 2025a,b].In this subsection, we will show that SC-hedge can also be used to characterize non-identifiability in C-DMGs over DMGs in specific conditions. First, let us recall some useful concepts to define a hedge.

**Definition 9** (C-component, Tian and Pearl [2002]). *Let $\mathcal{G}^* = (\mathbb{V}^*, \mathbb{E}^*)$ be a graph. A subset of vertices $\mathbb{V}_C^* \subseteq \mathbb{V}^*$ such that $\forall V_1^*, V_n^* \in \mathbb{V}_C^*$, $\exists V_1^*, \cdots, V_n^* \in \mathbb{V}^*$ with $\forall 1 \leq i < n$, $V_i^* \leftrightarrow V_{i+1}^*$ is called a C-component.*

**Definition 10** (C-forest, Shpitser and Pearl [2006]). *Let $\mathcal{G}^* = (\mathbb{V}^*, \mathbb{E}^*)$ be a graph. If $\mathcal{G}^*$ is acyclic, $\mathcal{G}^*$ is a forest (i.e., every of its vertices has at most one child), and $\mathcal{G}^*$ is a C-component then $\mathcal{G}^*$ is called a C-forest. The vertices which have no children are called roots and we say a C-forest is $\mathbb{R}^*$-rooted if it has roots $\mathbb{R}^* \subseteq \mathbb{V}^*$.*

**Definition 11** (Hedge, Shpitser and Pearl [2006, 2008]). *Consider a graph $\mathcal{G}^* = (\mathbb{V}^*, \mathbb{E}^*)$ and two disjoint sets of vertices $\mathbb{X}^*, \mathbb{Y}^* \subseteq \mathbb{V}^*$. Let $\mathcal{F} = \left(\mathbb{V}_{\mathcal{F}}^*, \mathbb{E}_{\mathcal{F}}^*\right)$ and $\mathcal{F}' = \left(\mathbb{V}_{\mathcal{F}'}^*, \mathbb{E}_{\mathcal{F}'}^*\right)$ be two $\mathbb{R}^*$-rooted*

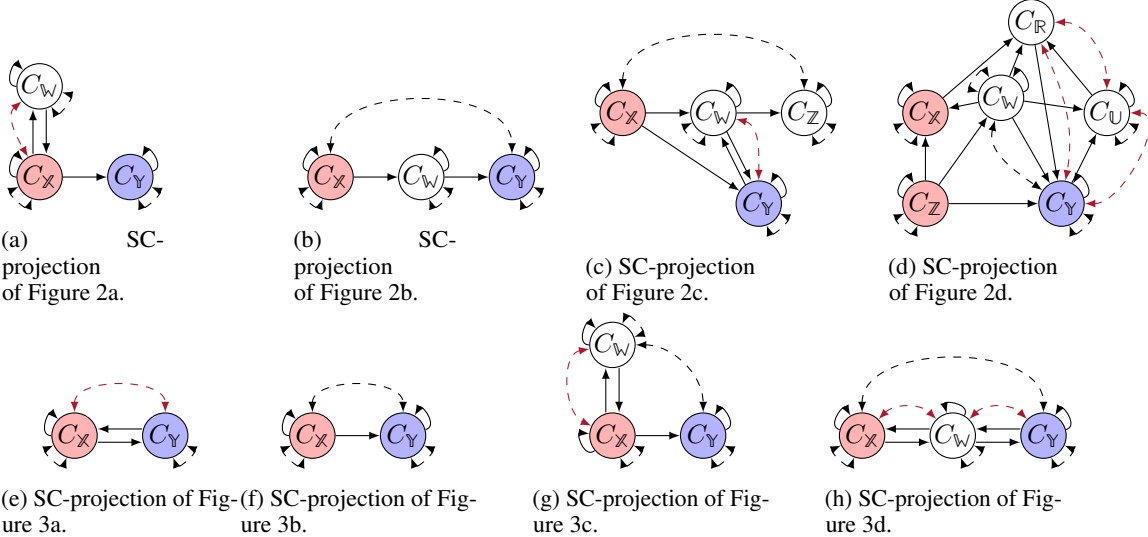

(a) SC-projection of Figure 2a.

(b) SC-projection of Figure 2b.

(c) SC-projection of Figure 2c.

(d) SC-projection of Figure 2d.

(e) SC-projection of Figure 3a.

(f) SC-projection of Figure 3b.

(g) SC-projection of Figure 3c.

(h) SC-projection of Figure 3d.

Figure 4: SC-projections of the C-DMGs in Figures 1, 2, and Figures 3. Each pair of red and blue vertices represents the total effect we are interested in, and the red edges indicate those added through the SC-projection.

*C-forests subgraphs of $\mathcal{G}^*$ such that $\mathbb{X}^* \cap \mathbb{V}^*_{\mathcal{F}} \neq \varnothing$, $\mathbb{X}^* \cap \mathbb{V}^*_{\mathcal{F}'} = \varnothing$, $\mathbb{F}' \subseteq \mathbb{F}$, and $\mathbb{R}^* \subset An\left(\mathbb{Y}^*, \mathcal{G}^* \backslash \mathbb{X}^*\right)$. Then $\mathbb{F}$ and $\mathbb{F}'$ form a hedge for the pair $\left(\mathbb{X}^*, \mathbb{Y}^*\right)$ in $\mathcal{G}^*$.*

As mentioned before, a hedge turned out to be too weak to cover non-identifiability in C-DMGs [Ferreira and Assaad, 2025b]. For example, the C-DMG in Figure 3a contains no hedge but the macro causal effect is not identifiable due to the cycle between $\mathbb{C}_{\mathbb{X}}$ and $\mathbb{C}_{\mathbb{Y}}$. In the following, we formally define SC-hedges in the context of C-DMGs over DMGs and demonstrate that this substructure serves as a sound criterion for detecting non-identifiable macro causal effects if every cluster in a cycle is of size strictly greater than 1.

**Definition 12** (Strongly connected projection (SC-projection)). *Consider a C-DMG $\mathcal{G}^{\mathbb{C}} = \left(\mathbb{C}, \mathbb{E}^{\mathbb{C}}\right)$. The SC-projection $\mathcal{H}^{\mathbb{C}}$ of $\mathcal{G}^{\mathbb{C}}$ is the graph that includes all vertices and edges from $\mathcal{G}^{\mathbb{C}}$, plus a dashed bidirected edge between each pair $C_{\mathbb{X}}, C_{\mathbb{Y}} \in \mathbb{C}$ such that $Scc\left(C_{\mathbb{X}}, \mathcal{G}^{\mathbb{C}}\right) = Scc\left(C_{\mathbb{Y}}, \mathcal{G}^{\mathbb{C}}\right)$ and $C_{\mathbb{X}} \neq C_{\mathbb{Y}}$.*

**Definition 13** (Strongly Connected Hedge (SC-Hedge)). *Consider an C-DMG $\mathcal{G}^{\mathbb{C}} = \left(\mathbb{C}, \mathbb{E}^{\mathbb{C}}\right)$, its SC-projection $\mathcal{H}^{\mathbb{C}}$ and two disjoint sets of vertices $\mathbb{C}_{\mathbb{X}}, \mathbb{C}_{\mathbb{Y}} \subseteq \mathbb{C}$. A hedge for $\left(\mathbb{C}_{\mathbb{X}}, \mathbb{C}_{\mathbb{Y}}\right)$ in $\mathcal{H}^{\mathbb{C}}$ is an SC-hedge for $\left(\mathbb{C}_{\mathbb{X}}, \mathbb{C}_{\mathbb{Y}}\right)$ in $\mathcal{G}^{\mathbb{C}}$.*

The following theorem guarantees the soundness of the SC-hedge criterion in C-DMGs over DMGs when every cluster in a cycle is of size strictly greater than 1. This assumption is useful as it allows the existence of compatible ADMGs even when the C-DMG over DMGs contains cycles. Thus, one can use Ferreira and Assaad 2025b, Theorem 5 in the presence of an SC-hedge to show that for every identifying sequence of do-calculus rules, there exists a compatible ADMG in which this sequence is not applicable.

**Theorem 5.** *Consider an C-DMG $\mathcal{G}^{\mathbb{C}} = \left(\mathbb{C}, \mathbb{E}^{\mathbb{C}}\right)$ such that every cluster which is in a cycle is of size at least 2 and two disjoint sets of vertices $\mathbb{C}_{\mathbb{X}}, \mathbb{C}_{\mathbb{Y}} \subseteq \mathbb{C}$. If there exists an SC-hedge for $\left(\mathbb{C}_{\mathbb{X}}, \mathbb{C}_{\mathbb{Y}}\right)$ in $\mathcal{G}^{\mathbb{C}}$ then $\Pr\left(\mathbb{c}_{\mathbb{y}} \mid do\left(\mathbb{c}_{\mathbb{x}}\right)\right)$ is not identifiable.*

The SC-projections of the C-DMGs illustrated in Figures 2 and 3 are given in Figure 4. One can notice that the SC-projections of the C-DMGs in Figure 3 all contain a hedge, thus the C-DMGs in Figure 3 contain a SC-hedge and the causal effect of interest is therefore not identifiable according to Theorem 5. In contrast, the projections of the C-DMGs in Figure 2 do not contain a hedge, this highlights the usefulness of SC-hedges for causal effect identification.

# 4 Conclusion

In this paper, we established the soundness and completeness of $\sigma$-separation and the do-calculus using $\sigma$-separation for identifying macro causal effects in C-DMGs over DMGs. There are two main limitations to this work. The first limitation is that the completeness result in Theorem 4 does not take into account that there might exist different sequences of rules of the do-calculus in different DMGs that can give the same final identification of the causal effect. Moreover, the completeness results only link the cluster graphs to the compatible underlying graphs, our results say nothing on the completeness of $\sigma$-separation and do-calculus in DMGs. A second related limitation is that we provided a graphical characterization for the non-identifiability of macro causal effects, however this characterization is not proven to be complete, even though we did not find any counter-example of its completeness. Proving it to be complete remains an open problem.

## Acknowledgments and Disclosure of Funding

This work was supported by the CIPHOD project (ANR-23-CPJ1-0212-01).

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

# A   Appendix

## A.1   Additional Notations and Properties

Firstly, the proofs require the definitions of intervened graphs and extended graphs which are the graphs induced respectively by intervened ioSCMs and extended ioSCMs Forré and Mooij [2020].

**Definition 14** (Intervened graph). *Let $\mathcal{G} = (\mathbb{V}, \mathbb{E})$ be a DMG induced from an ioSCM $\mathcal{M}$ and $\mathbb{A} \subseteq \mathbb{V}$ a set of variables. The intervened graph $\left(\mathcal{G}|_{do(\mathbb{A})}\right)$ is obtained by removing all edges going to $\mathbb{A}$ in $\mathcal{G}$.*

The intervened graph $\left(\mathcal{G}|_{do(\mathbb{A})}\right)$ is also written as $\mathcal{G}_{\overline{\mathbb{A}}}$ in [Pearl, 2009].

**Definition 15** (Extended graph). *Let $\mathcal{G} = (\mathbb{V}, \mathbb{E})$ be a DMG induced from an ioSCM $\mathcal{M}$. The extended graph $[\mathcal{G}] = ([\mathbb{V}], [\mathbb{E}])$ is obtained by adding for each vertex $V \in \mathbb{V}$ an extended vertex $I_V \in [\mathbb{V}]$ and the edge $I_V \to V \in [\mathbb{E}]$. i.e., $[\mathbb{V}] = \mathbb{V} \cup \{I_V \mid V \in \mathbb{V}\}$ and $[\mathbb{E}] = \mathbb{E} \cup \{I_V \to V \mid V \in \mathbb{V}\}$.*

In order to map the vertices in a C-DMG $\mathcal{G}^{\mathbb{c}}$ with the vertices in a compatible DMG $\mathcal{G}$, we will use the notion of corresponding cluster: $\forall C \in \mathbb{C}, \forall V \in C, Cl(V, \mathcal{G}^{\mathbb{c}}) = C$. Additionally, we will write for every set of clusters $\mathbb{C}_{\mathbb{A}} \subseteq \mathbb{C}, \mathbb{A} = \bigcup_{C \in \mathbb{C}_{\mathbb{A}}} C$.

**Definition 16** (Maximal compatible DMG). *Let $\mathcal{G}^{\mathbb{c}} = (\mathbb{C}, \mathbb{E}^{\mathbb{c}})$ be a C-DMG. Let us define the following sets:*

$$\mathbb{V} := \bigcup_{C \in \mathbb{C}} C$$
$$\mathbb{E}^{\to} := \{V \to V' \mid \forall V \in C, \ V' \in C' \text{ such that } C \to C' \in \mathbb{E}^{\mathbb{c}}\}$$
$$\mathbb{E}^{\leftrightarrow} := \{V \leftrightarrow V' \mid \forall V \in C, \ V' \in C' \text{ such that } C \leftrightarrow C' \in \mathbb{E}^{\mathbb{c}}\}$$
$$\mathbb{E} := \mathbb{E}^{\to} \cup \mathbb{E}^{\leftrightarrow}$$

*The graph $\mathcal{G}_m^{\mathbb{c}} = (\mathbb{V}, \mathbb{E})$ is compatible with $\mathcal{G}^{\mathbb{c}}$ and for every compatible DMG $\mathcal{G}$, we have $\mathcal{G} \subseteq \mathcal{G}_m^{\mathbb{c}}$, thus $\mathcal{G}_m^{\mathbb{c}}$ is called the maximal compatible DMG of $\mathcal{G}^{\mathbb{c}}$.*

**Property 1** (Compatibility of extended graphs). *Let $\mathcal{G}^{\mathbb{c}}$ be a C-DMG and $\mathcal{G}$ be a compatible DMG. Let us consider $[\mathcal{G}^{\mathbb{c}}]$ and $[\mathcal{G}]$ the corresponding extended graphs. Take $[\mathbb{V}] = \mathbb{V} \cup \{I_V \mid V \in \mathbb{V}\}$ the extended micro variables and $[\mathbb{C}] = \mathbb{C} \cup \{\{I_V \mid V \in C\} \mid C \in \mathbb{C}\}$ the extended partition. $[\mathcal{G}]$ is compatible with $[\mathcal{G}^{\mathbb{c}}]$ according to partition $[\mathbb{C}]$.*

*Proof.* Firstly, $\forall V, V' \in \mathbb{V}, V \to V' \in [\mathcal{G}]$ (resp. $\leftrightarrow$) $\iff V \to V' \in \mathcal{G}$ (resp. $\leftrightarrow$) $\iff Cl(V, \mathcal{G}^{\mathbb{c}}) \to Cl(V', \mathcal{G}^{\mathbb{c}}) \in \mathcal{G}^{\mathbb{c}}$ (resp. $\leftrightarrow$) $\iff Cl(V, \mathcal{G}^{\mathbb{c}}) \to Cl(V', \mathcal{G}^{\mathbb{c}}) \in [\mathcal{G}^{\mathbb{c}}]$ (resp. $\leftrightarrow$). Secondly, $\forall V \in \mathbb{V}, I_V \to V \in [\mathcal{G}]$ and $I_{Cl(V, \mathcal{G}^{\mathbb{c}})} \to Cl(V, \mathcal{G}^{\mathbb{c}}) \in [\mathcal{G}^{\mathbb{c}}]$. Lastly, $\forall C \in \mathbb{C}, I_C \to C \in [\mathcal{G}^{\mathbb{c}}]$ and $\exists V \in C, I_V \to V \in [\mathcal{G}]$ because $\mathbb{C}$ is a partition so $\forall C \in \mathbb{C}, C \neq \varnothing$. $\qquad\square$

**Property 2** (Extended maximal compatible graphs). *Let $\mathcal{G}^{\mathbb{c}}$ be a C-DMG, $\mathcal{G}_m^{\mathbb{c}}$ be the maximal compatible graph of $\mathcal{G}^{\mathbb{c}}$, $[\mathcal{G}_m^{\mathbb{c}}]$ be the extended graph of $\mathcal{G}_m^{\mathbb{c}}$ and $[\mathcal{G}^{\mathbb{c}}]_m$ be the maximal compatible graph of $[\mathcal{G}^{\mathbb{c}}]$. These graphs verify*

- *$[\mathcal{G}_m^{\mathbb{c}}] \subseteq [\mathcal{G}^{\mathbb{c}}]_m$, and*

- *$\forall V \in [\mathbb{V}], Scc(V, [\mathcal{G}_m^{\mathbb{c}}]) = Scc(V, [\mathcal{G}^{\mathbb{c}}]_m)$.*

*Therefore, any macro-level $\sigma$-connection that holds in $[\mathcal{G}^{\mathbb{c}}]_m$ also holds in $[\mathcal{G}_m^{\mathbb{c}}]$.*

*Proof.* Firstly, $\forall V, V' \in \mathbb{V}, V \to V' \in [\mathcal{G}_m^{\mathbb{c}}]$ (resp. $\leftrightarrow$) $\iff V \to V' \in \mathcal{G}_m^{\mathbb{c}}$ (resp. $\leftrightarrow$) $\iff Cl(V, \mathcal{G}^{\mathbb{c}}) \to Cl(V', \mathcal{G}^{\mathbb{c}}) \in \mathcal{G}^{\mathbb{c}}$ (resp. $\leftrightarrow$) $\iff Cl(V, \mathcal{G}^{\mathbb{c}}) \to Cl(V', \mathcal{G}^{\mathbb{c}}) \in [\mathcal{G}^{\mathbb{c}}]$ (resp. $\leftrightarrow$) $\iff V \to V' \in [\mathcal{G}^{\mathbb{c}}]_m$ (resp. $\leftrightarrow$). Secondly, $\forall V \in \mathbb{V}, I_V \to V \in [\mathcal{G}_m^{\mathbb{c}}]$ and $I_V \in Cl(I_V, [\mathcal{G}^{\mathbb{c}}]), V \in Cl(V, [\mathcal{G}^{\mathbb{c}}]), Cl(I_V, [\mathcal{G}^{\mathbb{c}}]) \to Cl(V, [\mathcal{G}^{\mathbb{c}}]) \in [\mathcal{G}^{\mathbb{c}}]$ thus $I_V \to V \in [\mathcal{G}^{\mathbb{c}}]_m$.

Regarding the strongly connected components, $\forall V \in \mathbb{V}, Scc(V, [\mathcal{G}_m^{\mathbb{c}}]) = Scc(V, \mathcal{G}_m^{\mathbb{c}}) = \{V\} \cup \left(\bigcup_{C \in \tilde{Scc}(Cl(V, \mathcal{G}^{\mathbb{c}}), \mathcal{G}^{\mathbb{c}})} C\right)^2 = \{V\} \cup \left(\bigcup_{C \in \tilde{Scc}(Cl(V, [\mathcal{G}^{\mathbb{c}}]), [\mathcal{G}^{\mathbb{c}}])} C\right) = Scc(V, [\mathcal{G}^{\mathbb{c}}]_m)$. Moreover, $\forall V \in$

---

[2]While the usual strongly connected component, $Scc(V^*, \mathcal{G}^*)$, always contains the vertex $V^*$ itself, this tweaked version, $\tilde{Scc}(V^*, \mathcal{G}^*)$, contains the vertex $V^*$ if and only if there exists a self-loop on it *i.e.*, $V^* \to V^* \in \mathcal{G}^*$.

$\mathbb{V}$, the set of edges including $I_V$ in $[\mathcal{G}_m^{\mathbb{c}}]$ is $\{I_V \to V\}$ and the set of edges including $I_V$ in $[\mathcal{G}^{\mathbb{c}}]_m$ is $\{I_V \to V' \mid V' \in Cl\,(V, \mathcal{G}^{\mathbb{c}})\}$ thus $Scc\,(I_V, [\mathcal{G}_m^{\mathbb{c}}]) = \{I_V\} = Scc\,(I_V, [\mathcal{G}^{\mathbb{c}}]_m)$.

In conclusion, using the Definition 8, any macro-level $\sigma$-connection that holds in $[\mathcal{G}^{\mathbb{c}}]_m$ also holds in $[\mathcal{G}_m^{\mathbb{c}}]$. $\qquad\square$

**Property 3** (Compatibility of intervened graphs)**.** *Let $\mathcal{G}^{\mathbb{c}}$ be a C-DMG and $\mathcal{G}$ be a compatible DMG. Let $\mathbb{C}_{\mathbb{A}} \subseteq \mathbb{C}$ be a set of clusters. Let us consider $\left(\mathcal{G}^{\mathbb{c}}|_{do(\mathbb{C}_{\mathbb{A}})}\right)$ and $\left(\mathcal{G}|_{do(\mathbb{A})}\right)$ the graphs obtained respectively by intervening on $\mathbb{C}_{\mathbb{A}}$ in $\mathcal{G}^{\mathbb{c}}$ and by intervening on $\mathbb{A}$ in $\mathcal{G}$. $\left(\mathcal{G}|_{do(\mathbb{A})}\right)$ is compatible with $\left(\mathcal{G}^{\mathbb{c}}|_{do(\mathbb{C}_{\mathbb{A}})}\right)$.*

*Moreover, let $\mathcal{G}_m^{\mathbb{c}}$ be the maximal compatible graph of $\mathcal{G}^{\mathbb{c}}$, $\left(\mathcal{G}_m^{\mathbb{c}}|_{do(\mathbb{A})}\right)$ its intervened graph and $\left(\mathcal{G}^{\mathbb{c}}|_{do(\mathbb{C}_{\mathbb{A}})}\right)_m$ be the maximal compatible graph of $\left(\mathcal{G}^{\mathbb{c}}|_{do(\mathbb{C}_{\mathbb{A}})}\right)$. $\left(\mathcal{G}_m^{\mathbb{c}}|_{do(\mathbb{A})}\right)$ and $\left(\mathcal{G}^{\mathbb{c}}|_{do(\mathbb{C}_{\mathbb{A}})}\right)_m$ are the same graph.*

*Proof.* $\forall V, V' \in \mathbb{V}$, $V \to V' \in \left(\mathcal{G}|_{do(\mathbb{A})}\right)$ (resp. $\leftarrow\negmedspace\rightarrow$) $\iff V \to V' \in \mathcal{G}$(resp. $\leftarrow\negmedspace\rightarrow$) and $V' \notin \mathbb{A} \iff Cl\,(V, \mathcal{G}^{\mathbb{c}}) \to Cl\,(V', \mathcal{G}^{\mathbb{c}}) \in \mathcal{G}^{\mathbb{c}}$(resp. $\leftarrow\negmedspace\rightarrow$) and $Cl\,(V', \mathcal{G}^{\mathbb{c}}) \notin \mathbb{C}_{\mathbb{A}} \iff Cl\,(V, \mathcal{G}^{\mathbb{c}}) \to Cl\,(V', \mathcal{G}^{\mathbb{c}}) \in \left(\mathcal{G}^{\mathbb{c}}|_{do(\mathbb{C}_{\mathbb{A}})}\right)$ (resp. $\leftarrow\negmedspace\rightarrow$). $\qquad\square$

**Property 4** (Intervened maximal compatible graphs)**.** *Let $\mathcal{G}^{\mathbb{c}}$ be a C-DMG, $\mathcal{G}_m^{\mathbb{c}}$ be the maximal compatible graph of $\mathcal{G}^{\mathbb{c}}$, $\left(\mathcal{G}_m^{\mathbb{c}}|_{do(\mathbb{A})}\right)$ be the intervened graph of $\mathcal{G}_m^{\mathbb{c}}$ and $\left(\mathcal{G}^{\mathbb{c}}|_{do(\mathbb{C}_{\mathbb{A}})}\right)_m$ be the maximal compatible graph of $\left(\mathcal{G}^{\mathbb{c}}|_{do(\mathbb{C}_{\mathbb{A}})}\right)$. $\left(\mathcal{G}_m^{\mathbb{c}}|_{do(\mathbb{A})}\right)$ and $\left(\mathcal{G}^{\mathbb{c}}|_{do(\mathbb{C}_{\mathbb{A}})}\right)_m$ are the same graph.*

*Proof.* $\forall V, V' \in \mathbb{V}$, $V \to V' \in \left(\mathcal{G}_m^{\mathbb{c}}|_{do(\mathbb{A})}\right)$ (resp. $\leftarrow\negmedspace\rightarrow$) $\iff V \to V' \in \mathcal{G}_m^{\mathbb{c}}$(resp. $\leftarrow\negmedspace\rightarrow$) and $V' \notin \mathbb{A} \iff Cl\,(V, \mathcal{G}^{\mathbb{c}}) \to Cl\,(V', \mathcal{G}^{\mathbb{c}}) \in \mathcal{G}^{\mathbb{c}}$(resp. $\leftarrow\negmedspace\rightarrow$) and $Cl\,(V', \mathcal{G}^{\mathbb{c}}) \notin \mathbb{C}_{\mathbb{A}} \iff Cl\,(V, \mathcal{G}^{\mathbb{c}}) \to Cl\,(V', \mathcal{G}^{\mathbb{c}}) \in \left(\mathcal{G}^{\mathbb{c}}|_{do(\mathbb{C}_{\mathbb{A}})}\right)$ (resp. $\leftarrow\negmedspace\rightarrow$) $\iff V \to V' \in \left(\mathcal{G}^{\mathbb{c}}|_{do(\mathbb{C}_{\mathbb{A}})}\right)_m$ (resp. $\leftarrow\negmedspace\rightarrow$). $\qquad\square$

## A.2 Proof of Theorem 1

*Proof.* Suppose $\mathbb{C}_{\mathbb{X}}$ and $\mathbb{C}_{\mathbb{Y}}$ are $\sigma$-separated by $\mathbb{C}_{\mathbb{W}}$ in $\mathcal{G}^{\mathbb{c}}$ and there exists a compatible DMG $\mathcal{G} = (\mathbb{V}, \mathbb{E})$ and a walk $\pi = \langle V_1, \cdots, V_n \rangle$ in $\mathcal{G}$ from $V_1 \in \mathbb{X}$ to $V_n \in \mathbb{Y}$ which is not $\sigma$-blocked by $\mathbb{W}$. Consider the walk $\tilde{\pi} = \langle C_1, \cdots, C_n \rangle$ with $\forall 1 \le i \le n, C_i = Cl\,(V_i, \mathcal{G}^{\mathbb{c}})$ and $\forall 1 \le i < n, \langle C_i \to C_{i+1} \rangle \subseteq \tilde{\pi}$ (resp. $\leftarrow, \leftarrow\negmedspace\rightarrow$) $\iff \langle V_i \to V_{i+1} \rangle \subseteq \pi$ (resp. $\leftarrow, \leftarrow\negmedspace\rightarrow$). $\tilde{\pi}$ is a walk from $\mathbb{C}_{\mathbb{X}}$ to $\mathbb{C}_{\mathbb{Y}}$ in $\mathcal{G}^{\mathbb{c}}$. Since $\mathbb{C}_{\mathbb{X}}$ and $\mathbb{C}_{\mathbb{Y}}$ are $\sigma$-separated by $\mathbb{C}_{\mathbb{W}}$, we know that $\mathbb{C}_{\mathbb{W}}$ $\sigma$-blocks $\tilde{\pi}$.

- If $C_1 \in \mathbb{W}$ or $C_n \in \mathbb{C}_{\mathbb{W}}$, then $V_1 \in \mathbb{W}$ or $V_n \in \mathbb{W}$ and thus $\pi$ is $\sigma$-blocked by $\mathbb{W}$ which contradicts the initial assumption.

Otherwise, take $1 < i < n$ such that $\langle C_{i-1}, C_i, C_{i+1} \rangle$ is $\mathbb{C}_{\mathbb{W}}$-$\sigma$-blocked.

- If $\langle C_{i-1} \leftarrow\negmedspace\rightarrow C_i \leftarrow\negmedspace\rightarrow C_{i+1} \rangle \subseteq \tilde{\pi}$ and $C_i \notin \mathbb{C}_{\mathbb{W}}$ then, $\langle V_{i-1} \leftarrow\negmedspace\rightarrow V_i \leftarrow\negmedspace\rightarrow V_{i+1} \rangle \subseteq \pi$ and $V_i \notin \mathbb{W}$. Thus $\pi$ is $\sigma$-blocked by $\mathbb{W}$ which contradicts the initial assumption.

- If $\langle C_{i-1} \leftarrow C_i \leftarrow\negmedspace\rightarrow C_{i+1} \rangle \subseteq \tilde{\pi}$ and $C_i \in \mathbb{C}_{\mathbb{W}} \backslash Scc\,(C_{i-1}, \mathcal{G}^{\mathbb{c}})$ then, $\langle V_{i-1} \leftarrow V_i \leftarrow\negmedspace\rightarrow V_{i+1} \rangle \subseteq \pi$. Moreover, $Scc\,(V_{i-1}, \mathcal{G}) \subseteq \bigcup_{C \in Scc(C_{i-1}, \mathcal{G}^{\mathbb{c}})} C$ and thus $V_i \in \mathbb{W} \backslash Scc\,(V_{i-1}, \mathcal{G})$. Therefore, $\pi$ is $\sigma$-blocked by $\mathbb{W}$ which contradicts the initial assumption.

- If $\langle C_{i-1} \leftarrow\negmedspace\rightarrow C_i \to C_{i+1} \rangle \subseteq \tilde{\pi}$ and $C_i \in \mathbb{C}_{\mathbb{W}} \backslash Scc\,(C_{i+1}, \mathcal{G}^{\mathbb{c}})$ then, $\langle V_{i-1} \leftarrow\negmedspace\rightarrow V_i \to V_{i+1} \rangle \subseteq \pi$. Moreover, $Scc\,(V_{i+1}, \mathcal{G}) \subseteq \bigcup_{C \in Scc(C_{i+1}, \mathcal{G}^{\mathbb{c}})} C$ and thus $V_i \in \mathbb{W} \backslash Scc\,(V_{i+1}, \mathcal{G})$. Therefore, $\pi$ is $\sigma$-blocked by $\mathbb{W}$ which contradicts the initial assumption.

- If $\langle C_{i-1} \leftarrow C_i \to C_{i+1} \rangle \subseteq \tilde{\pi}$ and $C_i \in \mathbb{C}_{\mathbb{W}} \backslash \left(Scc\,(C_{i-1}, \mathcal{G}^{\mathbb{c}}) \cap Scc\,(C_{i+1}, \mathcal{G}^{\mathbb{c}})\right)$ then, $\langle V_{i-1} \leftarrow V_i \to V_{i+1} \rangle \subseteq \pi$. Moreover, $\left(Scc\,(V_{i-1}, \mathcal{G}) \cap Scc\,(V_{i+1}, \mathcal{G})\right) \subseteq \bigcup_{C \in (Scc(C_{i-1}, \mathcal{G}^{\mathbb{c}}) \cap Scc(C_{i+1}, \mathcal{G}^{\mathbb{c}}))} C$ and thus $V_i \in \mathbb{W} \backslash \left(Scc\,(V_{i-1}, \mathcal{G}) \cap Scc\,(V_{i+1}, \mathcal{G})\right)$. Therefore, $\pi$ is $\sigma$-blocked by $\mathbb{W}$ which contradicts the initial assumption.

In conclusion, the $\sigma$-separation is sound in C-DMGs over DMGs.

$\qquad\square$

### A.3 Proof of Theorem 2

*Proof.* Suppose $\mathbb{C}_{\mathbb{X}}$ and $\mathbb{C}_{\mathbb{Y}}$ are not $\sigma$-separated by $\mathbb{C}_{\mathbb{W}}$ in $\mathcal{G}^{\mathbb{c}}$. There exists an $\mathbb{C}_{\mathbb{W}}$-$\sigma$-active path $\pi = \langle C_1, \cdots, C_n \rangle$ with $C_1 \in \mathbb{C}_{\mathbb{X}}$ and $C_n \in \mathbb{C}_{\mathbb{Y}}$. Take $\mathcal{G}_m^{\mathbb{c}} = (\mathbb{V}, \mathbb{E})$ the maximal compatible DMG of $\mathcal{G}^{\mathbb{c}}$ as in Definition 16. Take for every cluster $C \in \mathbb{C}$ a representative of this cluster $V_C \in \mathbb{C}$. The maximal compatible graph $\mathcal{G}_m^{\mathbb{c}}$ contains the path $\pi_m = \langle V_{C_1}, \cdots, V_{C_n} \rangle$ and for every cluster $C \in \mathbb{C}$ and every variable in that cluster $V \in C$, $Scc(V, \mathcal{G}_m^{\mathbb{c}}) = \bigcup_{C' \in Scc(C, \mathcal{G}^{\mathbb{c}})} C'$. Therefore, $\pi$ being $\mathbb{C}_{\mathbb{W}}$-$\sigma$-active in $\mathcal{G}^{\mathbb{c}}$ clearly implies that $\pi_m$ is $\mathbb{W}$-$\sigma$-active in $\mathcal{G}_m^{\mathbb{c}}$. In conclusion, the $\sigma$-separation criterion in C-DMGs over DMGs is complete.

Notice, that not only did we prove Theorem 2—*i.e.*,if a $\sigma$-separation does not hold in a C-DMG then there exists a compatible DMG in which the corresponding $\sigma$-separation does not hold—but we also explicitly exhibited this compatible DMG as being the maximal compatible DMG. $\qquad\square$

### A.4 Proof of Theorem 3

*Proof.* Let $\mathcal{G}^{\mathbb{c}} = (\mathbb{C}, \mathbb{E}^{\mathbb{c}})$ be a C-DMG, $\mathcal{G}$ a compatible DMG and $\mathbb{C}_{\mathbb{X}}, \mathbb{C}_{\mathbb{Y}}, \mathbb{C}_{\mathbb{Z}}, \mathbb{C}_{\mathbb{W}} \subseteq \mathbb{C}$ be disjoint subsets of vertices. Suppose a rule of the do-calculus applies in $\mathcal{G}^{\mathbb{c}}$ then Theorem 1, Property 1 and Property 3 guarantees that this rule applies in $\mathcal{G}$. More explicitly:

- If rule 1 applies *i.e.*, $(\mathbb{C}_{\mathbb{Y}} \perp\!\!\!\perp_\sigma \mathbb{C}_{\mathbb{X}} \mid \mathbb{C}_{\mathbb{W}}, \text{do}(\mathbb{C}_{\mathbb{Z}}))_{\mathcal{G}^{\mathbb{c}}}$, then using Theorem 1 as well as Properties 1 and 3 one knows that $(\mathbb{Y} \perp\!\!\!\perp_\sigma \mathbb{X} \mid \mathbb{W}, \text{do}(\mathbb{Z}))_{\mathcal{G}}$ and thus rule 1 applies in $\mathcal{G}$.

- If rule 2 applies *i.e.*, $(\mathbb{C}_{\mathbb{Y}} \perp\!\!\!\perp_\sigma \mathbb{I}_{\mathbb{C}_{\mathbb{X}}} \mid \mathbb{C}_{\mathbb{X}}, \mathbb{C}_{\mathbb{W}}, \text{do}(\mathbb{C}_{\mathbb{Z}}))_{\mathcal{G}^{\mathbb{c}}}$, then using Theorem 1 as well as Properties 1 and 3 one knows that $(\mathbb{Y} \perp\!\!\!\perp_\sigma \mathbb{I}_{\mathbb{X}} \mid \mathbb{X}, \mathbb{W}, \text{do}(\mathbb{Z}))_{\mathcal{G}}$ and thus rule 2 applies in $\mathcal{G}$.

- If rule 3 applies *i.e.*, $(\mathbb{C}_{\mathbb{Y}} \perp\!\!\!\perp_\sigma \mathbb{I}_{\mathbb{C}_{\mathbb{X}}} \mid \mathbb{C}_{\mathbb{W}}, \text{do}(\mathbb{C}_{\mathbb{Z}}))_{\mathcal{G}^{\mathbb{c}}}$, then using Theorem 1 as well as Properties 1 and 3 one knows that $(\mathbb{Y} \perp\!\!\!\perp_\sigma \mathbb{I}_{\mathbb{X}} \mid \mathbb{W}, \text{do}(\mathbb{Z}))_{\mathcal{G}}$ and thus rule 3 applies in $\mathcal{G}$.

Notice that because $\mathbb{C}_{\mathbb{X}}$ and $\mathbb{C}_{\mathbb{Z}}$ are disjoint, the actions of taking the intervened graph and taking the extended graph can be done in any order without any repercussion in the $\sigma$-separations of interest.

In conclusion, the do-calculus using $\sigma$-separation is sound in C-DMG over DMGs . $\qquad\square$

### A.5 Proof of Theorem 4

*Proof.* Let $\mathcal{G}^{\mathbb{c}} = (\mathbb{C}, \mathbb{E}^{\mathbb{c}})$ be a C-DMG, $\mathcal{G}_m^{\mathbb{c}}$ be the maximal compatible DMG and $\mathbb{C}_{\mathbb{X}}, \mathbb{C}_{\mathbb{Y}}, \mathbb{C}_{\mathbb{Z}}, \mathbb{C}_{\mathbb{W}} \subseteq \mathbb{C}$ be disjoint subsets of vertices. Suppose a rule of the do-calculus does not applies in $\mathcal{G}^{\mathbb{c}}$, then Theorem 2, Property 1 and Property 3 show that this rule does not apply in $\mathcal{G}_m^{\mathbb{c}}$. More explicitly:

- If rule 1 does not apply *i.e.*, $(\mathbb{C}_{\mathbb{Y}} \not\perp\!\!\!\perp_\sigma \mathbb{C}_{\mathbb{X}} \mid \mathbb{C}_{\mathbb{W}}, \text{do}(\mathbb{C}_{\mathbb{Z}}))_{\mathcal{G}^{\mathbb{c}}}$, then using Theorem 2 as well as Properties 2 and 4 one knows that $(\mathbb{Y} \not\perp\!\!\!\perp_\sigma \mathbb{X} \mid \mathbb{W}, \text{do}(\mathbb{Z}))_{\mathcal{G}_m^{\mathbb{c}}}$ and thus rule 1 does not apply in $\mathcal{G}_m^{\mathbb{c}}$.

- If rule 2 does not apply *i.e.*, $(\mathbb{C}_{\mathbb{Y}} \not\perp\!\!\!\perp_\sigma \mathbb{I}_{\mathbb{C}_{\mathbb{X}}} \mid \mathbb{C}_{\mathbb{X}}, \mathbb{C}_{\mathbb{W}}, \text{do}(\mathbb{C}_{\mathbb{Z}}))_{\mathcal{G}^{\mathbb{c}}}$, then using Theorem 2 as well as Properties 2 and 4 one knows that $(\mathbb{Y} \not\perp\!\!\!\perp_\sigma \mathbb{I}_{\mathbb{X}} \mid \mathbb{X}, \mathbb{W}, \text{do}(\mathbb{Z}))_{\mathcal{G}_m^{\mathbb{c}}}$ and thus rule 2 does not apply in $\mathcal{G}_m^{\mathbb{c}}$.

- If rule 3 does not apply *i.e.*, $(\mathbb{C}_{\mathbb{Y}} \not\perp\!\!\!\perp_\sigma \mathbb{I}_{\mathbb{C}_{\mathbb{X}}} \mid \mathbb{C}_{\mathbb{W}}, \text{do}(\mathbb{C}_{\mathbb{Z}}))_{\mathcal{G}^{\mathbb{c}}}$, then using Theorem 2 as well as Properties 2 and 4 one knows that $(\mathbb{Y} \not\perp\!\!\!\perp_\sigma \mathbb{I}_{\mathbb{X}} \mid \mathbb{W}, \text{do}(\mathbb{Z}))_{\mathcal{G}_m^{\mathbb{c}}}$ and thus rule 3 does not apply in $\mathcal{G}_m^{\mathbb{c}}$.

Notice that because $\mathbb{C}_{\mathbb{X}}$ and $\mathbb{C}_{\mathbb{Z}}$ are disjoint, considering the extended graph of the intervened graph or considering the intervened graph of the extended graph does not have any repercussion in the $\sigma$-connections of interest.

In conclusion, the do-calculus using $\sigma$-separation is complete in C-DMG over DMGs. $\qquad\square$

## A.6 Proof of Theorem 5

*Proof.* Let $\mathcal{G}^c = (\mathbb{C}, \mathbb{E}^c)$ be a C-DMG and take disjoint subsets $\mathbb{C}_{\mathbb{X}}, \mathbb{C}_{\mathbb{Y}} \subseteq \mathbb{C}$. Additionally, suppose that every cluster which is in a cycle in $\mathcal{G}^c$ is of size at least 2. More formally, $\forall C \in \mathbb{C}$, $|Scc(C, \mathcal{G}^c)| > 1 \implies |C| > 1$. Thanks to this assumption, one can view $\mathcal{G}^c$ as a C-DMG over ADMGs and thus use prior work[Ferreira and Assaad, 2025b]. Suppose there exists a SC-hedge for the pair $(\mathbb{C}_{\mathbb{X}}, \mathbb{C}_{\mathbb{Y}})$ in $\mathcal{G}^c$. Then, according to Theorem 5 of Ferreira and Assaad [2025b], the effect of $\mathbb{C}_{\mathbb{X}}$ on $\mathbb{C}_{\mathbb{Y}}$ is not identifiable.

In conclusion, the SC-hedge criterion is sound in C-DMG over DMGs under the additional assumption that every cluster which is in a cycle is of size at least 2. $\qquad\square$

