# OpenReview forum: "Identifying Macro Causal Effects in C-DMGs over DMGs"
_NeurIPS.cc/2025/Conference — NeurIPS 2025 poster_

### Official Review · Reviewer_6Zg1 · 2025-06-30

**Clarity:** 3
**Significance:** 3
**Originality:** 3
**Rating:** 5
**Confidence:** 3

**Summary:**

The paper tackles causal‐effect identification when the on a cluster-level abstraction called a C-DMG (nodes = variable clusters; edges = aggregate causal/confounding relations). Previous theory covered the acyclic case (ADMG); however, real systems often have feedback loops, motivating the cyclic DMG setting. The authors prove that, unlike ADMG setting, the do-calculus is unconditionally sound and complete for identifying macro causal effects in C-DMGs over DMGs. They also prove graphical criteria for non-identifiability of macro causal effects established in C-DMGs over ADMGs naturally extends to a subset of C-DMGs over DMGs. The work builds on input-output structural causal models (ioSCMs) which generalize SCMs to allow for cycles, providing a theoretical framework for causal identification in partially specified representations of cyclic systems.

**Questions:**

-I think in Theorem 3 it's better to recall what is c^w.

-Definition 7 is hard to follow without any explanation for each case.

-Have you considered developing an algorithm for identifying macro causal effects in C-DMGs over DMGs to help practitioners apply these results?

Could you elaborate on why clusters in cycles must have size greater than 1 for Theorem 5, and whether this requirement could be relaxed?

**Ethical Concerns:**

["NO or VERY MINOR ethics concerns only"]

**Final Justification:**

Thank you, authors, for the rebuttal. It answered most of my concerns. However, the manuscript should be updated as promised. I increased the quality score.

**Limitations:**

The authors adequately address the limitations of their work, particularly regarding the completeness of do-calculus and the soundness (but not proven completeness) of the SC-hedge criterion. They are transparent about the current boundaries of their theoretical results.
No guidance on how to learn C-DMGs from data, nor on the computational cost of the identification search. and no empirical validation or real-world case study.

**Paper Formatting Concerns:**

Nothing.

**Quality:**

4

**Strengths And Weaknesses:**

I found the paper’s narrative easy to follow and well organized. Although I had only passing familiarity with C-DMGs and ioSCMs, the exposition—especially the step-by-step illustrative examples—made the main ideas accessible. The identification / non-identification examples were particularly helpful. Overall, the work extends prior research in a meaningful direction and consistently provides the background needed to understand each new result. I should mention that I haven't read the proofs.

Weaknesses:

-The “completeness” theorem is relative: it connects C-DMGs to compatible underlying DMGs but does not establish absolute completeness for the cyclic setting, as the author acknowledged.

-The paper is entirely theoretical. No algorithmic implementation or empirical demonstration is offered. Can the authors design even a small synthetic experiment to showcase how the derived formulas recover interventional quantities in practice?

-The practical motivation for using C-DMGs over DMGs, rather than the acyclic C-DMGs over ADMGs studied earlier, is not fully articulated. Concrete real-world scenarios where true feedback cycles matter would strengthen the case for this extension.

-I think including intuition or proof sketches would be helpful in the main body.

---

> ### Author Rebuttal · Authors · 2025-07-29
>
> First of all, we would like to thank for the thorough review and interesting remarks. We will take into account the suggested modifications and carefully proofread the whole paper.
>
> As this is a fully theoretical paper, we are unsure as to what experiments could be relevant but we are willing to add some if the reviewer could detail what they had in mind.
> We are considering developing a package to help practitioners apply these results (and others) but this is beyond the scope of this paper.
>
> For real examples where feedback loops are important, we refer to the initial works on ioSCMs [Forre and Mooij 2020] where this is discusses more thoroughly.
> An conceptual example in the context of public health is the effect of a medication on behaviors:
> In the context of covid19, people who go party a lot are more likely to be contaminated so they will get vaccinated earlier to limit the risks, thus behaviors causes vaccinations.
> However, people who are vaccinated feel much safer and are more likely to go to a party, thus vaccination causes behaviors.
> To truly understand the effect of vaccination of contamination, one has to take this kind of feedback loops into account.
>
> Lastly regarding the necessity of the assumption for the hedge criterion: this assumption guarantees the existence of a compatible acyclic DMG in which one can use the classical hedge criterion.
> This is necessary as the $\sigma$-based do-calculus was not shown to be complete in DMGs and since the ID algorithm relies on the acyclicity assumption.
> To remove this assumption, one would first need to find an equivalent ID algorithm and/or a hedge criterion for DMGs.
> We will clarify this in the final version.
>
> Thanks again for the thoughtful comments. We hope to have answered all questions and we remain available during the discussion period in case there is any additional concerns.

---

> > ### Comment · Reviewer_6Zg1 · 2025-08-06
> >
> > Thank you for the rebuttal. It answered most of my concerns. However, the manuscript should be updated as promised.

---

### Official Review · Reviewer_rE2q · 2025-07-01

**Clarity:** 3
**Significance:** 3
**Originality:** 2
**Rating:** 4
**Confidence:** 3

**Summary:**

The paper proves soundness and completeness of the do calculus in a setting with macroscopic variables which are related by a possibly cyclic causal graph (resulting from coarse-graining a possibly cyclic graph with confounders into clusters of variables). The paper therefore extends known results for cluster graphs obtained from acyclic directed mixed graphs and generalizes them to cluster graphs obtained from directed mixed graphs.

**Questions:**

-	Please comment more on why generalization from C-DMG over ADMGs to C-DMG over DMGs raises serious challenges.

-	“However, Forré and Mooij [2020] showed that replacing d-separation by σ-separation in the three rules allows the do-calculus to be applicable on DMGs induced by ioSCMs.” I assume that it is crucial that the DMG is generated by an ioSCM because the semantics of interventions are not described a priori without specifying the generating model behind the DMG, right? Isn’t there just an ioSCM for the vector of cluster variables?

-	“which may contain true cyclic causal relations”: what is a non-true cyclic causla relation?


-	Inclomplete references: Philip Boeken and Joris M. Mooij. Dynamic structural causal models, 2024, and Patrick Forré and Joris M. Mooij. Constraint-based causal…


-	Abstract: “under the assumption that all clusters size are greater than 1.” Does the paper explain why this condition is necessary?

**Ethical Concerns:**

["NO or VERY MINOR ethics concerns only"]

**Final Justification:**

although one could argue the contribution to be incremental, it still is a nice and solid step forward.

**Limitations:**

I like that the discussion of limitations mentions the following subtlety: “The first limitation is that the completeness result in Theorem 4 does not take into account that there might exist different sequences of rules of the do-calculus in different DMGs that can give the same final identification of the causal effect”

A more serious limitation is my concern regarding relevance.

**Quality:**

3

**Strengths And Weaknesses:**

Strength:

The paper contains solid math, it is carefully written. The paper seems fair with respect to citing other work, it is very transparent about its contributions.


Weakness:

The relevance of aggregating variables into clusters of variables without dimensionality reduction is limited. Coarse-grained causal models in all applications I know work with aggregations that reduce the dimension, which raises problems that are conceptually harder than the problems solved here because interventions are no longer well-defined. I would appreciate a better motivation of clustering by convincing real-life examples.


Given that the results have already shown for C-DMG over ADMGs, extension to C-DMG over DMGs appears to incremental, given the concept of sigma-separation, which the paper builds upon.

I know that reviewers are often too fast with judging substantial work as incremental. Sometimes because results look straightforward in hindsight only and subtleties are no longer visible once the solution is there. The authors can try to convince me that I’m falling into the same fallacy by explaining subtleties and conceptual problems they needed to solve.

---

> ### Author Rebuttal · Authors · 2025-07-29
>
> First of all, we would like to thank for the thorough review and interesting remarks. We will take into account the mentioned typos and carefully proofread the whole paper.
>
> Concerning real examples of clustering without dimension reduction: this is common practice in different fields even though the soundness of this process was not guaranteed before [Anand et al 2023].
> In particular, cluster graphs (without dimension reduction) are very useful in health and epidemiology, for example [Piccininni et al 2023] uses the front-door criterion in an acyclic cluster graph where confounders are clustered and causal relationships between the confounders are not mentioned. Other examples can be found in [Anand and Hripcsak 2025].
> Moreover, cluster graphs without dimension reduction are also used for root cause analysis in IT monitoring [Assaad et al., 2023].
>
> Regarding the difference between [Ferreira and Assaad 2025] which focuses on C-DMGs over ADMGs and our work which focuses on C-DMGs over DMGs, a first difference is the use of $\sigma$-separation rather than d-separation as the underlying graphs might be cyclic.
> Moreover, the major difference in our opinion lies in the absence of assumption for the completeness results.
> Indeed [Ferreira and Assaad 2025] required that there is no two adjacent clusters of size 1 in a cycle to guarantee the completeness of d-separation and the completeness of do-calculus.
> In our work, this assumption is no longer required for the completeness results, it is only necessary for the soundness of the SC-hedge criterion. Removing this assumption is the most important aspect of the paper.
>
> Indeed we assume that that the underlying DMGs are induced by ioSCMs.
> It is also correct that a C-DMG over DMGs can be induced by a "macro"-ioSCM over the vector of cluster variables, just as a Cluster-DAG can be induced by a "macro"-SCM over the vector of cluster variables. However, the latter was not obvious until the seminal work of [Anand et al., 2023], which formally established this connection. In a similar way, we argue that the existence and formalization of a macro-level ioSCM over cluster variables was not evident prior to our work. In this respect, our contribution closely parallels that of [Anand et al., 2023], but in the context of ioSCMs.
>
> In C-DMG over DMGs there exists 2 distinct "types" of cycles. We call "true" cycles, cycles that appear in the true underlying ioSCM, in contrast to "false" cycles that appear only in the C-DMG because of the clustering process.
> For example if the true underlying DMG is $\mathbb{V}= ( X_1,Y_1,Y_2,Z_1 ) , \mathbb{E} = ( X_1\rightarrow Y_1, Y_1\rightarrow X_1, Y_1 \rightarrow Z_1, Z_1 \rightarrow Y_2 )$,
> and we cluster the variables according to the partition $C_\mathbb{X} = (X_1), C_\mathbb{Y} = (Y_1,Y_2), C_\mathbb{Z}= (Z_1)$,
> we get the C-DMG with edges $( C_\mathbb{X} \rightarrow C_\mathbb{Y}, C_\mathbb{Y} \rightarrow C_\mathbb{X}, C_\mathbb{Y} \rightarrow C_\mathbb{Z}, C_\mathbb{Z} \rightarrow C_\mathbb{Y})$.
> In this C-DMG, the cycle between $C_\mathbb{X}$ and $C_\mathbb{Y}$ is a "true" cycle and the cycle between $C_\mathbb{Y}$ and $C_\mathbb{Z}$ is a "false" cycle.
> We are willing to change the adjectives "true" and "false" if the reviewers believe it to be necessary and in any case we will clarify this in the final version.
>
> Lastly regarding the necessity of the assumption for the hedge criterion, we refer to our answer to reviewer 6Zg1 where this is discussed in details.
>
> Thanks again for the thoughtful comments. We hope to have answered all questions and we remain available during the discussion period in case there is any additional concerns.

---

> > ### Comment · Reviewer_rE2q · 2025-08-04
> > **convincing responses**
> >
> > I will raise my score

---

### Official Review · Reviewer_h1gx · 2025-07-03

**Clarity:** 3
**Significance:** 3
**Originality:** 3
**Rating:** 5
**Confidence:** 3

**Summary:**

Authors prove that do-calculus is sound and complete for C-DMGs over DMGs.

**Questions:**

- One of my major concerns is the actual instantiation of distributions for Example 4 and 5, especially given the presence of cycles in Example 5. In the introduction it is said that "cluster graphs allow for cycles, which can arise naturally in feedback systems or time-dependent processes, complicating the analysis compared to traditional ADMGs", but there is not explicit discussion on time-related effects, leaving the actual computation of the proposed identified effects dubious. I get the value of this contribution is to prove the correctness and completeness of the do-calculus for this class of models, but how can we use the output of the identification process in practice?

**Ethical Concerns:**

["NO or VERY MINOR ethics concerns only"]

**Final Justification:**

Authors addressed the issues I raised, the paper is particularly dense and it is not easy to follow due to the theoretical framework put in place, but with the additional explanations provided during the rebuttal everything is much more understandable.

**Limitations:**

- Discussed in the conclusions, but I would prefer to have a dedicated section to better highlight the current limits of the proposed approach.
- For instance, what are the actual implications of the statement: "there might exist different sequences of rules of the do-calculus in different DMGs that can give the same final identification of the causal effect"? Is it a limitation in the sense of "minimality" of sequence of rules application?

**Paper Formatting Concerns:**

- Using math blackboard font hinders readability, I would use bold or calligraphic bold fonts.
- The self loops/edges in Figures 1, subfigure c, node C_W (and following) are really hard to read/understand.

**Quality:**

3

**Strengths And Weaknesses:**

- Strengths
    - Proving the do-calculus to this class of models makes them useful to identify the so called "macro" causal effects.
- Weaknesses
    - Notation layers stratify one on top the other (e.g. E→ = E+ ∣ Vobs ∪ J, while there is an already defined V = Vobs ∪ J). I don't think the current notation choice helps the reader to follow the content of this dense paper.
    - Some references appears to be non-peer reviewed (e.g. either arXiv or "unpublished notes" references).

---

> ### Author Rebuttal · Authors · 2025-07-29
>
> First of all, we would like to thank for the thorough review and interesting remarks. We will take into account the suggested notation modifications and carefully proofread the whole paper.
>
> Concerning the question about time-related effects, the article [Ferreira and Assaad 2025] tackles similar questions in the context of time series.
> However, in this work we focus in settings where time is not specially of interest.
> In this setting, cycles occur for example when there are equilibriums: take a system, in which variables are linked through feedback cycles (X->Y and X<-Y), then we wait for an equilibrium before measuring the variables.
> In this setting, an intervention on X will affect Y and an intervention on Y will affect X, therefore, the cycle is necessary to fully express the causal relationships.
>
> Regarding the estimation process in cluster graphs: when using C-DMGs, one does not know all the causal relationships between micro variables however, one has access to all the data of micro-variables.
> Therefore, once the identification process outputs an estimand, one can estimate the causal effect exactly as one would estimate a multivariate causal effect in any classical setting described by [Pearl 2000].
>
> As for the limitations, we are willing to discuss them more in details.
>
> Lastly, the completeness of the do-calculus result has been discussed with reviewer 2YQb, we refer to this discussion for a reformulation of the explication that follows.
> The completeness result guarantees that if a rule $R_i$ of do-calculus does not apply in a C-DMG then, there exists a compatible DMG in which $R_i$ does not apply either.
> This result can trivially be extended to sets of rules of do-calculus: if a sequence of rules of do-calculus $\mathcal{R}$ is not applicable in a C-DMG, then there exists a rule $R_i$ which is not applicable in this C-DMG thus there exists a compatible DMG $\mathcal{G}$ in which $R_i$ is not applicable either and lastly the sequence of rules $\mathcal{R}$ which contains $R_i$ is not applicable to the DMG $\mathcal{G}$.
> However, one could imagine different sequences of rules that lead to the same estimand.
> For example, suppose the corresponding $\sigma$-separations apply, one could have $Pr(Y\mid do(X)) = Pr(Y)$ in a DMG $\mathcal{G}$ using rule 3 of do-calculus, and have in another DMG $\mathcal{G}'$ $Pr(Y\mid do(X)) = Pr(Y\mid X)= Pr(Y)$ using rules 2 and 1 of do-calculus.
> Therefore, the completeness result of do-calculus does not guarantee that if a causal effect is not identifiable in a C-DMG, there does not exist an estimand valid in every compatible DMG.
>
> Thanks again for the thoughtful comments. We hope to have answered all questions and we remain available during the discussion period in case there is any additional concerns.

---

> > ### Comment · Area_Chair_REJn · 2025-08-06
> > **Feedback?**
> >
> > Dear h1gx, could you please answer the rebuttal from the authors?

---

> > ### Comment · Reviewer_h1gx · 2025-08-06
> >
> > Dear authors, thank you for your rebuttal, after reading it carefully and re-reading the cited examples/sections with this explanation everything is much more understandable now. I will raise my clarity points.

---

### Official Review · Reviewer_2YQb · 2025-07-14

**Clarity:** 3
**Significance:** 3
**Originality:** 2
**Rating:** 4
**Confidence:** 3

**Summary:**

Causal identification from partially specified causal structures is a fundamental problem that is receiving increasing attention in recent years.  The paper introduces the problem of identification of macro causal effects from C-DMGs against compatible DMGs.  Below are the main takeaways from the paper:

1) $\sigma$-separation in C-DMGs implies corresponding $\sigma$-separation in all compatible DMGs.

2) Non-$\sigma$-separation in C-DMGs implies existence of a compatible DMG where the corresponding variables are non-$\sigma$- separated.

3) The do-calculus (with $\sigma$ separation) is sound for C-DMGs.

4) Non-applicability of a do-calculus rule implies non-applicability of the same rule in some compatible DMG.

5) An application of SC-Hedge for C-DMGs over DMGs, for a restricted class of graphs.

**Questions:**

The notion of `completeness of do-calculus' originated from the early works of Tian and Pearl, which is different from the one discussed here.  I would suggest clarifying the definition of completeness over DMGs earlier in the manuscript, possibly even in the abstract.

I am trying to understand if the non-applicability of a set of do-calculus rules $\mathcal{R}$ on a given C-DMG would imply existence of a single compatible DMG where none of the rules in $\mathcal{R}$ are applicable?  Such a claim, if true would make Theorem 4 much stronger.

Below are some minor suggestions / typos:

1. I suggest using Pa, An, De, Scc and other abbreviations in non-math mode.

2. Figure 1:  Two ADMGs → Two DMGs

3. Some typos in lines 160 – 164 (due to its, lead to cycle between clusters)

4. Please define compatible graphs and extended graphs – the core components of the proof.

5. Line 228: in in

6. Line 266:  conditioned → intervened

7. Please proofread Appendix.

**Ethical Concerns:**

["NO or VERY MINOR ethics concerns only"]

**Final Justification:**

I am happy with the rebuttal and the author discussions.  With the addition of the discussed corollary, I believe the submission makes substantial contribution to causal identification from partial structures.  I would like to maintain my score.

**Limitations:**

yes

**Paper Formatting Concerns:**

nil

**Quality:**

4

**Strengths And Weaknesses:**

Strengths:  The manuscript is well-written and I really enjoyed reading the paper with nice running examples.  The technical contributions appear sound and advance the state of the art in causal identification from partial causal graphs.

Weaknesses:  The theorem statements are nice, my only issue is that the proofs are relatively straightforward and most of the concepts extend from previous work on ADMGs.  Proving one of the following results: do-calculus completeness for non-identifiability, or a more general graphical condition for non-identiability would have greatly improved the contributions from a technical/novelty standpoint.  The proofs in Appendix are hard to understand and they rely on two important concepts: the extended graphs and compatible graphs, whose formal definitions are missing in the manuscript.

---

> ### Author Rebuttal · Authors · 2025-07-29
>
> First of all, we would like to thank for the thorough review and interesting remarks. We will take into account the typos mentioned and carefully proofread the whole paper.
>
> Indeed, for the do-calculus, our notion of completeness is different from that of Tian and Pearl as we only guarantee that if an effect is not identifiable in a C-DMG then for every identifying sequence of rules of do-calculus $\mathcal{R}$, there exists a compatible underlying DMG in which $\mathcal{R}$ is not applicable, or in other words there exists a rule $R_i \in \mathcal{R}$ and a compatible DMG $\mathcal{G}$ for which $R_i$ is not applicable in $\mathcal{G}$. However, it may very well be that for every individual rule $R_i$ in $\mathcal{R}$ there exists a compatible graph in which $R_i$ is applicable. We will make sure to clarify this point in the final version.
>
> The notions of extended graphs and intervened graphs can be found in [Forre and Mooij 2020]. However, as we use those notions in the proofs in appendix, we will properly state the definitions at the beginning of the appendix. Moreover, we will make explicit what we mean by "compatible" graph.
>
> Thanks again for the thoughtful comments. We hope to have answered all questions and we remain available during the discussion period in case there is any additional concerns.

---

> > ### Comment · Reviewer_2YQb · 2025-08-04
> >
> > Dear Authors,  Thank you for your rebuttal.  I am just curious to understand if my question on the improvement of Theorem~4 holds.

---

> > > ### Author Response · Authors · 2025-08-04
> > >
> > > Dear Reviewer, you suggested to improve Theorem 4 (completeness of do-calculus) to "if a set of do-calculus rules $\\mathcal{R}$ is not applicable in a given C-DMG then there exists a compatible DMG in which none of the rules are applicable".
> > > However, we do not believe this to be true.
> > > Firstly, let us make more explicit the notion of "non-applicability": a set $\\mathcal{R}$ of rules of do-calculus is not applicable in a C-DMG $\\mathcal{G}$, if there exists a rule $R\\in\\mathcal{R}$ which is not applicable in $\\mathcal{G}$.
> > > Therefore, a set of rules $\\mathcal{R}$ may very well be non-applicable in a C-DMG, even if most rules in $\\mathcal{R}$ are applicable and only a single rule $R\\in \\mathcal{R}$ is not applicable.
> > > With this in mind, one can build a simple counterexample of the suggested improvement of the theorem by taking the C-DMG $\\mathcal{G} = \\left\\{C\_{X} \\rightarrow C\_{Y}\\right\\}$ and $\\mathcal{R} = \\left\\{ \\text{rule 3: } C\_{Y} {\\perp\\!\\!\\!\\perp}\_{\\mathcal{G}\_{\\overline{C\_{X}}}} C\_{X} ; \\text{rule 3: } C\_{X} {\\perp\\!\\!\\!\\perp}\_{\\mathcal{G}\_{\\overline{C\_{Y}}}} C\_{Y}  \\right\\}$. In this example, the set of rules $\\mathcal{R}$ is not applicable in $\\mathcal{G}$ as the first $\\sigma$-separation statement "$C\_{Y} {\\perp\\!\\!\\!\\perp}\_{\\mathcal{G}\_{\\overline{C\_{X}}}} C\_{X}$" is not true, however in every DMG compatible with $\\mathcal{G}$, the second $\\sigma$-separation statement "$C\_{X} {\\perp\\!\\!\\!\\perp}\_{\\mathcal{G}\_{\\overline{C\_{Y}}}} C\_{Y}$" is always true.
> > >
> > > To go further, if by your suggestion, you meant "if $\\mathcal{R}$ is a set of do-calculus rules such that $\\forall R \\in \\mathcal{R}$, $R$ is not applicable in the C-DMG $\\mathcal{G}$, then there exists a compatible DMG in which none of the rules are applicable" then the answer is a bit less forward.
> > > We do believe this statement to be true, moreover, one can exhibit this compatible DMG as it is the densest compatible DMG $\\mathcal{G}\_m$.
> > > In fact the proof of Theorem 4 actually builds this densest compatible DMG $\\mathcal{G}\_m$ and shows that the rules of do-calculus which are not applicable in the C-DMG $\\mathcal{G}$ are not applicable in the densest compatible graph $\\mathcal{G}\_m$.
> > > Moreover, if one takes for granted Theorem 4 as stated "If one of the do-calculus rules does not apply for a given C-DMG over DMGs $\\mathcal{G}$, then there exists a compatible DMG for which the corresponding rule does not apply" and the well known fact that if a $\\sigma$-separation holds in a graph, then it holds in every subgraph in which edges were removed, then one can prove your suggested theorem as every DMG compatible with a C-DMG $\\mathcal{G}$ is a subgraph of the densest compatible DMG $\\mathcal{G}\_m$.
> > >
> > > We hope this clarifies our work and if not, we are willing to discuss it more.

---

> > > > ### Comment · Reviewer_2YQb · 2025-08-04
> > > >
> > > > Dear Authors,
> > > >
> > > > Thank you for your response, I understand it now.  Indeed, my question was "if $\mathcal{R}$ is a set of do-calculus rules such that $\forall R \in \mathcal{R}$, $R$ is not applicable in the C-DMG $\mathcal{G}$, then there exists a compatible DMG in which none of the rules are applicable".  I suggest to add this as a corallary after Theorem~4 along with a brief explanation.
> > > >
> > > > I would like to maintain my score.

---

> > > > > ### Author Response · Authors · 2025-08-05
> > > > >
> > > > > Thank you for the suggestion, it is a great idea. We will add this corollary in the final version.

---

### Note · Authors · 2025-08-15

We would like to use this opportunity to express once more our gratitude towards the reviewers for their valuable feedback and the constructive discussions that followed.

---

### Decision · Program_Chairs · 2025-09-17

**Decision:**

Accept (poster)

**Comment:**

This paper presents a theoretically grounded extension of causal identification techniques using cyclic directed mixed graphs. The manuscript is well-written and accessible, even to readers with limited familiarity with the underlying graphical frameworks, thanks to clear exposition and helpful examples. The main contribution lies in extending do-calculus to this class of models, which enables the identification of macro-level causal effects in settings that include feedback cycles.
However, the novelty and technical depth of the work are questioned. While the theorem statements are sound, the proofs are seen as relatively straightforward extensions of existing results on acyclic directed mixed graphs. The lack of formal definitions for key concepts like extended and compatible graphs in the appendix further weakens the rigor. Additionally, the paper does not offer a completeness result for non-identifiability or a broader graphical condition that could have significantly strengthened its theoretical contribution.
The notation used throughout the paper is dense and sometimes confusing, which may hinder readability. Some references are non-peer-reviewed, and the paper lacks empirical validation. Although the theoretical results are promising, the absence of algorithmic implementation or synthetic experiments makes it difficult to assess their practical utility. Including even a small-scale demonstration of how the derived formulas recover interventional quantities would enhance the paper’s impact.
The reviewers also raised the issue that the authors do not sufficiently explain why feedback cycles are essential in real-world scenarios or how their framework addresses challenges that arise in such settings. The relevance of clustering variables without dimensionality reduction is also questioned, as most coarse-grained models in practice involve dimensionality changes that complicate intervention definitions.
Despite these limitations, the paper is appreciated for its clarity, transparency, and solid mathematical foundation. It contributes meaningfully to the theoretical landscape of causal inference, but its significance would be clearer with stronger motivation, deeper theoretical results, and some empirical grounding.
However, rebuttals and subsequent discussion that was engaing, addressed and somewhat solved many issue raised by the reviews.